# GRAPH EDIT NETWORKS

**Benjamin Paassen**
The University of Sydney
benjamin.paassen@sydney.edu.au

**Daniele Grattarola**
Università della Svizzera italiana
daniele.grattarola@usi.ch

**Daniele Zambon**
Università della Svizzera italiana
daniele.zambon@usi.ch

**Cesare Alippi**
Università della Svizzera italiana
Politecnico di Milano
cesare.alippi@usi.ch

**Barbara Hammer**
Bielefeld University
bhammer@techfak.uni-bielefeld.de

## ABSTRACT

While graph neural networks have made impressive progress in classification and regression, few approaches to date perform time series prediction on graphs, and those that do are mostly limited to edge changes. We suggest that graph edits are a more natural interface for graph-to-graph learning. In particular, graph edits are general enough to describe any graph-to-graph change, not only edge changes; they are sparse, making them easier to understand for humans and more efficient computationally; and they are local, avoiding the need for pooling layers in graph neural networks. In this paper, we propose a novel output layer - the graph edit network - which takes node embeddings as input and generates a sequence of graph edits that transform the input graph to the output graph. We prove that a mapping between the node sets of two graphs is sufficient to construct training data for a graph edit network and that an optimal mapping yields edit scripts that are almost as short as the graph edit distance between the graphs. We further provide a proof-of-concept empirical evaluation on several graph dynamical systems, which are difficult to learn for baselines from the literature.

## 1 INTRODUCTION

Recent advances in graph representation learning have mostly focused on tasks of classification or regression, *i.e.* tasks with graph-structured input but numeric output (Battaglia et al., 2018; Kipf & Welling, 2016a; Veličković et al., 2018). By contrast, few approaches to date can transform a graph-structured input to a graph-structured output (Hajiramezanali et al., 2019; Paaßen et al., 2018; Zambon et al., 2019). This lacuna is crucial because time series prediction on graphs requires graph-structured output, namely the next graph in a time series. Applications of time series prediction on graphs include epidemiological models (Keeling & Eames, 2005), social (Liben-Nowell & Kleinberg, 2007; Masuda & Holme, 2019), telecommunications (Nanavati et al., 2006), traffic (Cui et al., 2019), citation (Shibata et al., 2012), and financial transaction networks (Chan & Olmsted, 2017), as well as student solutions in intelligent tutoring systems (Paaßen et al., 2018). In each of these settings, predicting the changes in graphs can deepen the understanding of the domain and provide useful knowledge for designing interventions.

Currently, methods for time series prediction on graphs are limited to the dynamics of the node attributes (Yu et al., 2018), or changes in connectivity (Goyal et al., 2020; Hajiramezanali et al., 2019), but do not cover changes in the node set. Fortunately, there exists a rich research tradition of edit distances (e.g. Levenshtein, 1965; Zhang & Shasha, 1989; Sanfeliu & Fu, 1983) which can describe *any* change between two graphs. Further, edits are sparse and have a simple semantic (delete, insert, relabel), which makes them easier to interpret for human observers and makes them computationally

more efficient (linear instead of quadratic) compared to a dense representation. Finally, edits are local, enabling us to make edit decisions at each node instead of coordinating information across the entire graph.

In this work, we connect graph neural networks to edit distances by developing a simple, linear output layer that maps node embeddings to graph edits. We call our output layer the graph edit network (GEN). We also develop a general training and inference scheme to transform any graph $G_t$ to its successor $G_{t+1}$ using only local binary edit decisions and a regression for node attributes.

Theoretically, we prove that a) a mapping between the nodes of $G_t$ and $G_{t+1}$ is sufficient to construct training data for the GEN, b) this construction yields almost no overhead compared to directly transforming the mapping to graph edits, and c) provided that the mapping between $G_t$ and $G_{t+1}$ is optimal and the GEN can perfectly reproduce the training data, the edit script is almost as short as the graph edit distance (Sanfeliu & Fu, 1983).

In addition to this core theoretical contribution, we provide a proof-of-concept of our model by demonstrating that GENs can learn a variety of dynamical systems on graphs which are more difficult to handle for baseline systems from the literature. We also show that the sparsity of edits enables GENs to scale up to realistic graphs with thousands of nodes.

## 2 BACKGROUND

**Graph Neural Networks:** Graph neural networks (GNNs) compute representations of nodes in a graph by aggregating information of neighboring nodes (Bacciu et al., 2020; Defferrard et al., 2016; Kipf & Welling, 2016a; Micheli, 2009; Scarselli et al., 2009). In particular, the representation $\phi^l(v) \in \mathbb{R}^{n_l}$ of node $v$ in layer $l$ is computed as follows:

$$\phi^l(v) = f^l_{\text{merge}}\Big(\phi^{l-1}(v), f^l_{\text{aggr}}\big(\{\phi^{l-1}(u)|u \in \mathcal{N}(v)\}\big)\Big) \tag{1}$$

where $\mathcal{N}(v)$ is some neighborhood of $v$ in the graph and $f^l_{\text{merge}}$ as well as $f^l_{\text{aggr}}$ are functions that aggregate the information of their arguments, returning a single vector (Xu et al., 2019). The representation in the 0th layer is usually defined as the initial node attributes or a constant vector for all nodes (Kipf & Welling, 2016a). Recently, many implementations of $f^l_{\text{merge}}$, $f^l_{\text{aggr}}$, and neighborhood $\mathcal{N}$ have been suggested, such as a weighted sum via the graph Laplacian (Kipf & Welling, 2016a), recurrent neural networks (Hamilton et al., 2017), or attention mechanisms (Veličković et al., 2018). Our approach is agnostic to the choice of graph neural network. We merely require some vectorial embedding for each node in the input graph.

**Graph Generators:** Multiple works in recent years have proposed recurrent models to generate graphs (Bacciu et al., 2019; Li et al., 2018; You et al., 2018a;b; Zhang et al., 2019). Roughly speaking, these recurrent models first output a node, and then all connections of this node to previous nodes until a special end-of-sentence token is produced. While such a scheme does enable time series prediction, it only works for insertions, *i.e.* starting at an empty graph and inserting nodes and edges over time. If one wishes to account for general graph changes, one first has to encode a graph into a vector and then decode from this vector the graph in the next time step, similar to molecular design (Jin et al., 2019; Fu et al., 2020). However, such models have to generate the next graph from scratch and can not exploit the sparsity and interpretability of edits, as we suggest.

**Link Prediction:** Link prediction is complementary to graph generation. It assumes a constant number of nodes, but changing connectivity between them (Liben-Nowell & Kleinberg, 2007; Richard et al., 2014; Shibata et al., 2012). Typical link prediction approaches compute node features first, followed by an affinity index between nodes based on their features. Finally, edges with low index are predicted to vanish, while edges with high index are predicted to appear. For example, Goyal et al. (2020) combine dense and recurrent blocks to build an autoencoder for link prediction, while Hajiramezanali et al. (2019) combine a GNN and a RNN to obtain a spatio-temporal variational graph autoencoder. In GENs, we predict edge changes with a similar scheme, using a graph neural network to obtain the node features and then mapping these node features to the graph changes. However, in contrast to prior work, we do not predict the next adjacency matrix but only the change in adjacencies, which is a much sparser signal, reducing the time complexity from quadratic to linear. Additionally, GENs can not only handle edge changes, but also node changes.

We note that our current paper is limited to a Markovian setting, *i.e.* we do not consider the past for computing node representations. This limitation could be addressed by combining our output layer with EvolveGCN (Pareja et al., 2020) which uses a recurrent net to predict the weights of a graph neural net, thus being able to handle changes in the node set.

**Dynamic Attributes:** Recently, graph neural networks have been extended to predict changes to node attributes, while the nodes and edges remain fixed (Cui et al., 2019; Seo et al., 2018), which is particularly useful for traffic networks. GENs are complementary to these works, in that we consider the more general case of graph topology changes.

**Time Series Prediction on Graphs:** To our knowledge, only very few works to date have addressed the most general case of time series of graphs, where both nodes and edges are permitted to change. In particular, Paaßen et al. (2018) suggest several kernel-based time series prediction methods for graphs. However, their scheme is limited to predictions in the kernel space and mapping a prediction back to a graph requires solving an inverse kernel problem, relying on approximations that impact accuracy (Paaßen et al., 2018). Zambon et al. (2019) embed the time series into a vector space using a GNN and use a recurrent neural network to predict the next time step. To obtain the corresponding graph, a multi-layer perceptron is used to compute the adjacency matrix and node features from the predicted embedding. Besides being computationally expensive, this dense decoder also assumes a fixed order of the nodes.

**Graph Edits:** The basis for our approach are graph edits, which are functions that describe changes in graphs (Sanfeliu & Fu, 1983). Formally, we first define an attributed, directed graph as a triple $G = (V, E, \boldsymbol{X})$, where $V = \{1, \ldots, N\}$ is a finite set of node indices, $E \subseteq V \times V$ is a set of edges, and $\boldsymbol{X} \in \mathbb{R}^{N \times n}$ is a matrix of node attributes for some $n \in \mathbb{N}$. We define the nodes as indices for notational simplicity, but we do not assume any specific order, *i.e.* we treat isomorphic graphs as the same. Now, let $\mathcal{G}$ be the set of all possible attributed directed graphs. We define a *graph edit* as some function $\delta : \mathcal{G} \to \mathcal{G}$. In particular, we consider the graph edits of Sanfeliu & Fu (1983), namely *node deletions* $\mathrm{del}_i$, which delete the $i$th node from a graph, *node replacements* $\mathrm{rep}_{i,x}$, which set the attribute of node $i$ to $x$, *node insertions* $\mathrm{ins}_x$, which add a new node with attribute $x$ to a graph, *edge deletions* $\mathrm{edel}_{i,j}$, which delete the edge $(i, j)$ from a graph, and *edge insertions* $\mathrm{eins}_{i,j}$, which insert the edge $(i, j)$ into a graph. We then define an *edit script* $\bar{\delta}$ as a finite sequence $\bar{\delta} = \delta_1, \ldots, \delta_T$ of graph edits and we define the application of $\bar{\delta}$ as the composition of all edits, *i.e.* $\bar{\delta}(G) := \delta_T \circ \ldots \circ \delta_1(G)$.

Finally, we define the graph edit distance $d_{\mathrm{GED}}(G, G')$ between two graphs $G$ and $G'$ as the length of the shortest script $\bar{\delta}$ such that $\bar{\delta}(G) \cong G'$, where $\cong$ means isomorphic. The GED is well-defined and a proper metric, *i.e.* a script connecting any two graphs always exists, the GED between two isomorphic graphs is zero, the GED is symmetric, and it conforms to the triangular inequality (Abu-Aisheh et al., 2015; Sanfeliu & Fu, 1983). While prior work has already attempted to approximate the graph edit distance with graph neural nets (Bai et al., 2019; Li et al., 2019) our work is, to our knowledge, the first to produce actual graph edits as network output, and to avoid graph pooling layers.

## 3  GRAPH EDIT NETWORKS

Let $G_1, G_2, \ldots, G_T$ be a time series of graphs. Our goal is to develop a neural network that takes a graph $G_t$ as input and outputs graph edits that transform $G_t$ to $G_{t+1}$. For simplicity we make a Markov assumption, *i.e.* $G_t$ is assumed to be sufficient to predict $G_{t+1}$ (future research could address this limitation, e.g. by applying EvolveGCN of Pareja et al., 2020).

Now, let $G_t = (V, E, \boldsymbol{X})$ be an attributed graph with $N$ nodes. Our proposed processing pipeline has three steps. First, we use some graph neural network (refer to Equation 1) to compute a matrix of node embeddings $\boldsymbol{\Phi} \in \mathbb{R}^{N \times n}$. Second, we use a linear layer to compute numerical edit scores that express which nodes and edges should be deleted, inserted, and relabeled, respectively. Third, we translate these scores via Algorithm 1 to an edit script $\bar{\delta}$ and apply this script to the input graph to obtain the output graph $\bar{\delta}(G_t)$. This pipeline is also illustrated in Figure 1.

In the remainder of this section, we describe the graph edit network layer (Section 3.1), our training scheme (Section 3.2), our inference scheme (Section 3.3), and finally our theoretical results (Section 3.4).

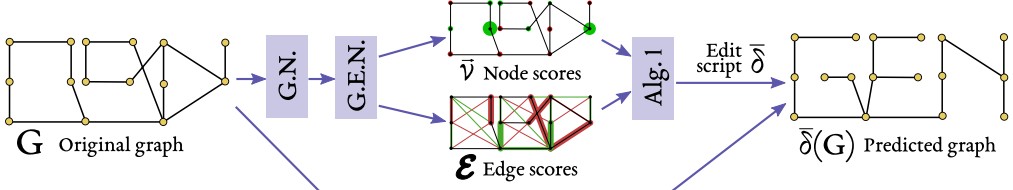

Figure 1: An illustration of the processing pipeline. An input graph $G$ is processed by a message passing network (Equation 1). The output layer is a GEN, and produces node scores $\vec{\nu}$ and edge scores $\mathcal{E}$. Algorithm 1 translates the node and edge scores into an edit script $\bar{\delta}$ which, when applied to the input graph $G$, constructs the predicted graph $\bar{\delta}(G)$.

### 3.1 GRAPH EDIT NETWORK LAYER

Our proposed graph edit network (GEN) is a linear layer to compute edit scores that express which nodes and edges should be deleted, inserted, or relabeled. The input of our GEN is a matrix $\mathbf{\Phi} \in \mathbb{R}^{N \times n}$ of node embeddings as returned by a graph neural network (refer to Equation 1). We then compute node edit scores $\vec{\nu} \in \mathbb{R}^N$, edge filter scores $\vec{e}^+ \in \mathbb{R}^N$ as well as $\vec{e}^- \in \mathbb{R}^N$, and new node attributes $Y$ via linear maps from $\mathbf{\Phi}$. After this is done, we consider only those pairs $(i, j)$ where $e_i^+ > 0$ and $e_j^- > 0$ and compute an edge edit score $\epsilon_{i,j}$ via another linear layer that receives $\vec{\phi}_i, \vec{\phi}_j$, and the inner product $\vec{\phi}_i^T \cdot \vec{\phi}_j$ as inputs. The interpretation of these scores is that $\nu_i$ should be positive if a new node connected to $i$ is inserted, $\nu_i$ should be negative if node $i$ is deleted, $e_i^+, e_j^-$, as well as $\epsilon_{i,j}$ should be positive if edge $(i, j)$ is inserted and $e_i^+, e_j^-$, as well as $-\epsilon_{i,j}$ should be positive if edge $(i, j)$ is deleted. Note that we only compute the edge edit score $\epsilon_{i,j}$ for edges $(i, j)$ where $e_i^+ > 0$ and $e_j^- > 0$. Thus, if edge changes concern only a number of nodes in $\mathcal{O}(\sqrt{n})$, the GEN layer operates in linear instead of quadratic time. We can also enforce the linear time by setting all $e_i^+$ and $e_j^-$ to zero that are not in the top $R$ for some $R$ that is either constant or in $\mathcal{O}(\sqrt{n})$.

**From scores to edits:** Next, we translate these scores into edits. The formal translation scheme is given in Algorithm 1. Roughly speaking, we delete any node $i$ where the node edit score $\nu_i$ is smaller than $-\frac{1}{2}$, we insert a new node with attribute $\vec{y}_i$, connected to $i$, whenever $\nu_i$ is larger than $+\frac{1}{2}$, and we replace the attribute $\vec{y}_i$ with $\vec{y}_i$ otherwise. For edges, we delete any edge $(i, j)$ where $\epsilon_{i,j} < -\frac{1}{2}$, where $\nu_i < -\frac{1}{2}$, or where $\nu_j < -\frac{1}{2}$, and we insert any edge where $\epsilon_{i,j} > +\frac{1}{2}$. The complexity of Algorithm 1 is as follows. In line 2, we first construct $|E \cap \{(i,j)|\nu_i < -\frac{1}{2} \text{ or } \nu_j < -\frac{1}{2}\}|$ edits, which is bounded by $|E|$, which in turn is in $\mathcal{O}(N)$ for a sparse graph and $\mathcal{O}(N^2)$ for a dense graph. Lines 3-8 perform $|\{(i,j)|e_i^+ > 0, e_j^- > 0\}|$ iterations, which can be bounded to some constant $R^2$ by the edge filtering trick above. Lines 9-15 iterate over all nodes several times, which is in $\mathcal{O}(N)$. The space complexity is the same since we add one edit each iteration, which needs to be stored. The overall time and space complexity is thus $\mathcal{O}(N)$ for sparse graphs and $\mathcal{O}(N^2)$ for dense graphs.

Note the special case of node insertions in this scheme. As with other edits, we make the decision to insert a new node locally at each node instead of globally for the graph. This relieves the need to aggregate information across the entire graph. Further, by connecting a new node directly with an existing one, we ensure that any two new nodes can be distinguished purely based on their graph connectivity, without relying on auxiliary information.

### 3.2 TRAINING

The key challenge in training GENs is to identify which scores the network *should* produce such that the GEN transforms the input graph $G_t$ into its desired successor $G_{t+1}$. In other words, we require a *teaching signal* consisting of ground truth scores $(\hat{\nu}, \hat{Y}, \hat{e}^+, \hat{e}^-, \hat{\mathcal{E}})$, such that the edit script $\bar{\delta}$ returned by Algorithm 1 yields $\bar{\delta}(G_t) \cong G_{t+1}$.

Unfortunately, such a one-step teaching signal is sometimes insufficient. Consider the two example graphs $G_t = (\{1\}, \emptyset, (0))$ and $G_{t+1} = (\{1, 2, 3\}, \{(1, 2), (1, 3)\}, (0, 0, 0)^T)$. In this case, there exists no one-step teaching signal that transforms $G_t$ to $G_{t+1}$ because we can only insert

---

**Algorithm 1** The scheme to translate the outputs of the GEN layer $\nu_i$, $\vec{y}_i$, $e_i^+$, $e_i^-$, and $\epsilon_{i,j}$ to graph edits.

---

1: **function** TRANSLATE(graph $G = (V, E, \boldsymbol{X})$, node edit scores $\vec{\nu} \in \mathbb{R}^N$, attributes $\boldsymbol{Y} \in \mathbb{R}^{N \times n}$, edge filter scores $\vec{e}^+, \vec{e}^- \in \mathbb{R}^N$, and edge edit scores $\boldsymbol{\mathcal{E}} \in \mathbb{R}^{N \times N}$)
2:     Initialize script $\bar{\delta}$ with $\text{edel}_{i,j}$ for all $(i,j)$ with $\nu_i < -\frac{1}{2}$ or $\nu_j < -\frac{1}{2}$ in lexicographic order.
3:     **for** $i$ with $e_i^+ > 0$ and $\nu_i \geq -\frac{1}{2}$ **do**
4:         **for** $j$ with $e_j^- > 0$ and $\nu_j \geq -\frac{1}{2}$ **do**
5:             Append $\text{edel}_{i,j}$ to $\bar{\delta}$ if $(i,j) \in E$ and $\epsilon_{i,j} < -\frac{1}{2}$.
6:             Append $\text{eins}_{i,j}$ to $\bar{\delta}$ if $(i,j) \notin E$ and $\epsilon_{i,j} > +\frac{1}{2}$.
7:         **end for**
8:     **end for**
9:     Append $\text{rep}_{i,\vec{y}_i}$ to $\bar{\delta}$ for all $i$ with $|\nu_i| \leq \frac{1}{2}$ and $\vec{x}_i \neq \vec{y}_i$.
10:     $k \leftarrow 1$.
11:     **for** $i$ with $\nu_i > +\frac{1}{2}$ **do**
12:         Append $\text{ins}_{\vec{y}_i}$ and $\text{eins}_{i,N+k}$ to $\bar{\delta}$.
13:         $k \leftarrow k + 1$.
14:     **end for**
15:     Append $\text{del}_i$ to $\bar{\delta}$ for all $i$ with $\nu_i < -\frac{1}{2}$ in descending order.
16:     **return** $\bar{\delta}$.
17: **end function**

---

as many new nodes as already exist. However, it is possible to set up a two-step teaching signal $(\hat{\nu}_1, \hat{\boldsymbol{Y}}_1, \hat{e}_1^+, \hat{e}_1^-, \hat{\boldsymbol{\mathcal{E}}}_1), (\hat{\nu}_2, \hat{\boldsymbol{Y}}_2, \hat{e}_2^+, \hat{e}_2^-, \hat{\boldsymbol{\mathcal{E}}}_2)$ with $\hat{\nu}_1 = (1)$, $\hat{\nu}_2 = (1, 0)^T$, and all other scores set to zero. When plugging these values into Algorithm 1 we obtain the edits $\bar{\delta}_1 = \text{ins}, \text{eins}_{1,2}$ and $\bar{\delta}_2 = \text{ins}, \text{eins}_{1,3}$, such that the concatenation $\bar{\delta} = \bar{\delta}_1, \bar{\delta}_2$ does indeed yield $\bar{\delta}(G_t) \cong G_{t+1}$. In general, Theorem 2 (below) shows that a teaching signal with $K + 1$ steps suffices to transform any input graph $G_t$ into any output graph $G_{t+1}$, where $K$ is the number of insertions necessary to transform $G_t$ into $G_{t+1}$ and the first $K$ steps only perform these insertions.

Provided that a teaching signal exists, our training procedure should ensure that the actual edit scores $(\vec{\nu}_1, \boldsymbol{Y}_1, \vec{e}_1^+, \vec{e}_1^-, \boldsymbol{\mathcal{E}}_1), \ldots (\vec{\nu}_{K+1}, \boldsymbol{Y}_{K+1}, \vec{e}_{K+1}^+, \vec{e}_{K+1}^-, \boldsymbol{\mathcal{E}}_{K+1})$ result in the same edits as the teaching signal when being plugged into Algorithm 1. To do so we treat every edit decision in Algorithm 1 as a binary classification and punish different decisions with a classification loss, such as the hinge loss (Zhao et al., 2017) or crossentropy (refer to Appendices B and C for a more details on loss functions).

In particular, in the first $K$ steps of a teaching signal, we have a classification loss for the decision of inserting a node ($\nu_{k,i} > +\frac{1}{2}$) or not ($\nu_{k,i} \leq +\frac{1}{2}$), plus a loss for punishing deviations between predicted attributes $\vec{y}_{k,i}$ and desired attributes $\hat{y}_{k,i}$ for all $i$ with $\hat{\nu}_{k,i} > +\frac{1}{2}$. For step $K + 1$, the same idea applies. For each node we have a binary classification loss for the decision of deleting a node $i$ ($\nu_{K+1,i} < -\frac{1}{2}$) or not; plus the regression loss between $\vec{y}_{K+1,i}$ and $\hat{y}_{K+1,i}$ for non-deleted nodes $i$; plus the classification loss regarding whether to change the outgoing edges of node $i$ ($e_{K+1,i}^+ > 0$) or not; plus the classification loss regarding whether to change the incoming edges of node $i$ ($e_{K+1,i}^- > 0$) or not; plus the classification loss regarding whether to delete an existing edge $(i,j)$ ($\epsilon_{i,j} < -1$) or not; plus the classification loss regarding whether to insert a non-existing edge $(i,j)$ ($\epsilon_{i,j} > 1$) or not. The sum of all these losses forms our training loss for a single graph pair $(G_t, G_{t+1})$. Because this loss is differentiable, we can train our neural net end-to-end by performing a gradient descent scheme on this loss, e.g. using Adam (Kingma & Ba, 2015).

Importantly, edge filtering needs to be adjusted to the training data. In particular, if we impose a limit on the maximum number of nodes with $e_i^+ > 0$ or $e_i^- > 0$, this limit should be higher than the maximum number of those nodes in the teaching signals for the training data - otherwise, the training can never achieve zero loss. If the limit is high enough, however, the training procedure is the same with or without the limit. The limit only impacts inference.

### 3.3 INFERENCE

Once training is complete, we wish to use our GEN for inference. In particular, given a previously unseen graph $G_t$, we want to know its successor $G_{t+1}$. Because $G_{t+1}$ is unknown, we can not construct a teaching signal. Instead, we plug the current graph $G_t$ into our graph neural net to compute node features $\mathbf{\Phi}$, then use the GEN to compute scores $(\vec{\nu}, \mathbf{Y}, \vec{e}^+, \vec{e}^-, \mathbf{\mathcal{E}})$, and plug these into Algorithm 1 to retrieve an edit script $\bar{\delta}$, which then yields our predicted graph $G_{t+1} = \bar{\delta}(G_t)$ (also refer to Figure 1). This is the simple one-step scheme that we will also use in our graph dynamical system experiments.

In case our training data requires multi-step teaching signals, our inference also needs multiple steps. In particular, we use the following scheme: 1) Color all nodes red. 2) If no red nodes are left, go to 4. Otherwise, re-compute the node features $\vec{\phi}_i$, the node edit score $\nu_i$, and the attributes $\vec{y}_i$ for all red nodes $i$. 3) For all $i$ with $\nu_i > +\frac{1}{2}$, insert a new node with attributes $\vec{y}_i$, draw an edge from $i$ to the new node, and color the new node red. Color all nodes $i$ with $\nu_i \leq +\frac{1}{2}$ blue. Then go to 2.
4) Compute the node features $\mathbf{\Phi}$ and the edit scores $(\vec{\nu}, \mathbf{Y}, \vec{e}^+, \vec{e}^-, \mathbf{\mathcal{E}})$ for all nodes. Set all $\nu_i > \frac{1}{2}$ to zero. Then, call Algorithm 1 and apply the resulting script to the graph.

In general, it may be necessary to provide state information (e.g. via a recurrent neural net) to ensure that the network can distinguish whether it still needs to insert nodes or not. In our experiments, however, it is sufficient to either summarize all edits in a single step or to supply the network with a binary node attribute that flags whether we are still in 'insertion mode' or not.

### 3.4 THEORY

In this section, we wish to answer two questions. First, how to construct a teaching signal for a graph edit network? Second, how short can the output edit script of a GEN get, provided that it still transforms graph $G_t$ to graph $G_{t+1}$?

The answer to both questions lies in making the theoretical connection to graph edit distances more explicit. To do so, we first introduce a key concept that will help us, namely that of a graph mapping.

**Definition 1** (Graph Mapping). Let $G = (V = \{1, \ldots, M\}, E, \mathbf{X})$ and $G' = (V' = \{1, \ldots, N\}, E', \mathbf{X}')$ be two non-empty graphs, *i.e.* $M, N > 0$. Then, we define a *graph mapping* $\psi$ between $G$ and $G'$ as a bijective mapping $\psi : \{1, \ldots, M + N\} \rightarrow \{1, \ldots, M + N\}$ with the additional restriction that for the set $\mathrm{Ins}_\psi := \{j \leq N | \psi^{-1}(j) > M\}$ we obtain $\psi^{-1}(\mathrm{Ins}_\psi) = \{M + 1, \ldots, M + |\mathrm{Ins}_\psi|\}$.

Graph mappings are useful because they are intimately connected to edit scripts. In particular, we re-state a result from the literature that any edit script converts to a graph mapping and back and that this conversion never increases the length of the script.

**Theorem 1** (Script to mapping). *Let $G$ and $G'$ be any two graphs. There exist two polynomial algorithms to translate any edit script $\bar{\delta}$ with $\bar{\delta}(G) \cong G'$ to a graph mapping $\psi_{\bar{\delta}}$ between $G$ and $G'$, and to translate any graph mapping $\psi$ between $G$ and $G'$ into an edit script $\bar{\delta}_\psi$ with $\bar{\delta}_\psi(G) \cong G'$, such that for any $\bar{\delta}$ with $\bar{\delta}(G) \cong G'$ we obtain $|\bar{\delta}_{\psi_{\bar{\delta}}}| \leq |\bar{\delta}|$.*

One proof is contained in Bougleux et al. (2017). We provide a more extensive version in Appendix A.1, which also gives more details on the structure of $\psi_{\bar{\delta}}$ and $\bar{\delta}_{\psi_{\bar{\delta}}}$.

Next, we show that a graph mapping can also be used to construct a teaching signal for a GEN. Even better, the conversion from mapping to teaching signal to edit script yields almost as short scripts as the direct conversion from mappings to edit scripts. Indeed, we can provide a sharp bound for the overhead in terms of the connected components in the target graph.

**Theorem 2** (Mapping to teaching signal). *Let $G$ and $G'$ be any two graphs with $M$ and $N$ nodes, respectively. There exists an $\mathcal{O}(M^2 + N^2)$ algorithm (namely Algorithm 2 in the appendix) that translates any graph mapping $\psi$ with $|\mathrm{Ins}_\psi| < N$ between $G$ and $G'$ into a teaching signal $(\hat{\nu}_1, \hat{\mathbf{Y}}_1, \hat{e}_1^+, \hat{e}_1^-, \hat{\mathbf{\mathcal{E}}}_1), \ldots, (\hat{\nu}_{K+1}, \hat{\mathbf{Y}}_{K+1}, \hat{e}_{K+1}^+, \hat{e}_{K+1}^-, \hat{\mathbf{\mathcal{E}}}_{K+1})$ such that the output of Algorithm 1 is a script $\bar{\delta}$, where the following holds: 1) $\bar{\delta}(G) \cong G'$; 2) $|\bar{\delta}| \leq |\bar{\delta}_\psi| + 2 \cdot (C - 1)$, where $C$ is the number of connected components in $G'$ (this bound is sharp); 3) the first $K$ steps contain only*

*insertions, with $\nu_{k,i} = 0 \Rightarrow \nu_{k+1,i} = \ldots = \nu_{K+1,i} = 0$; 4) the last step contains no insertions; 5) $K \leq |\mathrm{Ins}_\psi|$ (this bound is sharp).*

Refer to Appendix A.2 for the proof. So this result tells us that we obtain a (sparse) teaching signal if we have a good graph mapping $\psi$. But how to obtain $\psi$? In many cases, we can exploit domain knowledge. For example, if we know that the input graphs are trees, we can use the polynomial tree edit distance algorithm to infer mappings that correspond to short edit scripts (Zhang & Shasha, 1989); or, if node IDs are given (like user IDs in social networks), we can set $\psi$ such that it maintains IDs. In our experiments, we follow such domain-specific schemes to construct the mappings $\psi$.

But there is also a general strategy connected to the graph edit distance. While computing the graph edit distance itself is NP-hard (Bougleux et al., 2017), one can achieve a good approximation by constructing a graph mapping via the Hungarian algorithm and then converting this mapping to an edit script via Theorem 1 (Riesen & Bunke, 2009; Abu-Aisheh et al., 2017; Blumenthal et al., 2020). For our purposes, we can simply apply such an approximator and then use the graph mapping to construct our teaching signal. Importantly, if the approximator happens to find the optimal mapping and if our graph edit network is powerful enough to achieve a zero loss on one graph tuple $(G_t, G_{t+1})$, then the output of our graph edit network is close to the graph edit distance.

**Corollary 1** (Near-Optimality of Graph Edit Network Architecture). *Let $G$ and $G'$ be any two non-empty graphs and let $C$ be the number of connected components in $G'$. Further, let $\psi$ be a graph mapping between $G$ and $G'$ such that $|\bar\delta_\psi| = d_{GED}(G_t, G_{t+1})$. Finally, let $f$ be a graph edit network that reproduces the teaching signal, i.e. $f(G) = \bar\delta$ with $\bar\delta$ being the same script as the result of Theorem 2 for $\psi$. Then, it holds: $|\bar\delta| \leq d_{GED}(G, G') + 2 \cdot (C - 1)$.*

As mentioned before, this corollary only holds if the mapping is optimal and if the GEN can reproduce the teaching signal. The latter only holds if the node features $\Phi$ are rich enough to make each edit classification problem linearly separable (because any edit decision is a linear binary classification). Typically, this fails if a graph neural net can not distinguish two nodes that would need to be treated differently. For example, in an unlabeled ring graph $G = (\{1, \ldots, N\}, \{(1, 2), \ldots, (N, 1)\})$, Equation 1 assigns the same node embedding to all nodes and, hence, the GEN returns the same edits. In such cases, distinguishing information must be integrated via an alternative architecture or via node attributes (which is the strategy we take in the experiments). For further work on the expressiveness of graph neural nets we point the reader to Xu et al. (2019).

To summarize: The proposed way to use a GEN is 1) to gather a training time series of graphs $G_1, \ldots, G_{T+1}$; 2) to set up reference graph mappings $\psi_t$ between $G_t$ and $G_{t+1}$ for all $t \in \{1, \ldots, T\}$, e.g. via graph edit distance approximators; 3) to compute teaching signals via Theorem 2; 4) to initialize an appropriately powerful graph neural net with a final GEN layer and train it to reproduce the teaching signals on the training data; 5) to use the trained GEN in inference.

## 4 EXPERIMENTS

Our experimental evaluation displays the capability of GENs on a set of graph dynamical systems in comparison to baselines from the literature. Experiments are reported in three groups and cover graph dynamical systems, tree dynamical systems, and a social network dataset. All experiments require the possibility of changing nodes, and almost all require additional edge changes. The data is discussed in more detail in Appendix D. We perform all experiments on a consumer grade laptop with core i7 CPU. All experimental code is available at `https://gitlab.com/bpaassen/graph-edit-networks`.

First, we consider the following three graph dynamical systems.

**Edit Cycles:** A manually defined dataset of cycles in the set of undirected graphs with up to four nodes. The teaching protocol is hand-crafted to perform optimal edits between each graph and its successor. To sample a time series we let the cycle run for 4-12 time steps at random. The node features $\phi^0(x)$ were set to zero.

**Degree Rules:** A dynamical system on undirected graphs of arbitrary size with the following rules. First, delete every node with a degree larger than 3. Second, connect nodes that share at least one common neighbor. Third, insert a new node at any node with a degree lower than 3. We used the

Figure 2: Two graph time series from the *degree rules* dataset. Blue arrows indicate graph dynamics, labelled with the teaching signal.

Table 1: The average precision and recall values ($\pm$ std.) across five repeats for all edit types on the graph dynamical systems.

| model | node insertion | | node deletion | | edge insertion | | edge deletion | |
|---|---|---|---|---|---|---|---|---|
| | recall | precision | recall | precision | recall | precision | recall | precision |
| | | | | | edit cycles | | | | |
| VGAE | $0.62 \pm 0.0$ | $1.00 \pm 0.0$ | $1.00 \pm 0.0$ | $0.69 \pm 0.1$ | $1.00 \pm 0.0$ | $1.00 \pm 0.0$ | $1.00 \pm 0.0$ | $1.00 \pm 0.0$ |
| VGRNN | $0.64 \pm 0.0$ | $1.00 \pm 0.0$ | $0.63 \pm 0.0$ | $1.00 \pm 0.0$ | $0.95 \pm 0.0$ | $0.06 \pm 0.0$ | $1.00 \pm 0.0$ | $0.71 \pm 0.1$ |
| XE-GEN | $1.00 \pm 0.0$ | $1.00 \pm 0.0$ | $1.00 \pm 0.0$ | $1.00 \pm 0.0$ | $1.00 \pm 0.0$ | $1.00 \pm 0.0$ | $1.00 \pm 0.0$ | $1.00 \pm 0.0$ |
| GEN | $1.00 \pm 0.0$ | $1.00 \pm 0.0$ | $1.00 \pm 0.0$ | $1.00 \pm 0.0$ | $1.00 \pm 0.0$ | $1.00 \pm 0.0$ | $1.00 \pm 0.0$ | $1.00 \pm 0.0$ |
| | | | | | degree rules | | | | |
| VGAE | $0.15 \pm 0.0$ | $1.00 \pm 0.0$ | $1.00 \pm 0.0$ | $0.96 \pm 0.0$ | $0.88 \pm 0.0$ | $0.97 \pm 0.1$ | $1.00 \pm 0.0$ | $0.97 \pm 0.1$ |
| VGRNN | $0.14 \pm 0.0$ | $1.00 \pm 0.0$ | $0.72 \pm 0.0$ | $1.00 \pm 0.0$ | $0.56 \pm 0.0$ | $0.21 \pm 0.0$ | $1.00 \pm 0.0$ | $0.02 \pm 0.0$ |
| XE-GEN | $1.00 \pm 0.0$ | $1.00 \pm 0.0$ | $1.00 \pm 0.0$ | $1.00 \pm 0.0$ | $0.97 \pm 0.0$ | $0.99 \pm 0.0$ | $1.00 \pm 0.0$ | $1.00 \pm 0.0$ |
| GEN | $1.00 \pm 0.0$ | $1.00 \pm 0.0$ | $1.00 \pm 0.0$ | $1.00 \pm 0.0$ | $0.97 \pm 0.1$ | $0.99 \pm 0.0$ | $1.00 \pm 0.0$ | $1.00 \pm 0.0$ |
| | | | | | game of life | | | | |
| VGAE | $0.27 \pm 0.1$ | $1.00 \pm 0.0$ | $1.00 \pm 0.0$ | $0.03 \pm 0.0$ | $1.00 \pm 0.0$ | $1.00 \pm 0.0$ | $1.00 \pm 0.0$ | $1.00 \pm 0.0$ |
| VGRNN | $0.31 \pm 0.1$ | $1.00 \pm 0.0$ | $0.32 \pm 0.1$ | $1.00 \pm 0.0$ | $1.00 \pm 0.0$ | $0.00 \pm 0.0$ | $1.00 \pm 0.0$ | $0.01 \pm 0.0$ |
| XE-GEN | $1.00 \pm 0.0$ | $1.00 \pm 0.0$ | $1.00 \pm 0.0$ | $0.98 \pm 0.0$ | $1.00 \pm 0.0$ | $1.00 \pm 0.0$ | $1.00 \pm 0.0$ | $1.00 \pm 0.0$ |
| GEN | $1.00 \pm 0.0$ | $1.00 \pm 0.0$ | $1.00 \pm 0.0$ | $1.00 \pm 0.0$ | $1.00 \pm 0.0$ | $1.00 \pm 0.0$ | $1.00 \pm 0.0$ | $1.00 \pm 0.0$ |

node index in one-hot coding as node features $\phi^0(x)$. For every connected component in the input graph, this dynamical system provably converges to a 4-clique. We started with a random undirected adjacency matrix of size $8 \times 8$ and let the system run until convergence. Refer to Figure 2 for two example time series.

**Game of Life:** We simulated one of five oscillatory shapes in Conway's game of life (Gardner, 1970), namely blinker, glider, beacon, toad, and clock, for 10 time steps, placed on a random location on a $10 \times 10$ grid and additionally activated $10\%$ of the grid cells at random. We represented the grid as a graph with 100 nodes and represented the 8-neighborhood via the adjacency matrix. The desired edits were node deletions for all nodes that should switch from alive to dead and node insertions for all nodes that should switch from dead to alive. As node features we used the alive-state of the node.

Note that a single-step teaching signal is sufficient in all cases. We compare our graph edit network against variational graph autoencoders (VGAE) and variational graph recurrent nets (VGRNN) which predict the adjacency matrix of the next graph via an outer product of the node features, similar to our edge prediction scheme (Kipf & Welling, 2016b; Hajiramezanali et al., 2019). Note that neither net can predict node changes directly, but we predict a node deletion whenever all edges of a node are deleted. To train our GEN model, we apply both the hinge loss (GEN; Appendix B) as well as the crossentropy loss (XE-GEN; Appendix C). For all models, we use two graph neural network layers with 64 neurons each, sum as aggregation function and concatenation as merge function (refer to Equation 1). We train all networks with an Adam optimizer in pyTorch using a learning rate of $10^{-3}$ and stopping training after 30,000 time series or if the loss dropped below $10^{-3}$. After training, we evaluated the predictive performance on 10 additional time series. We repeated each experiment five times.

The recall and precision for all edit types, all models, and all datasets is shown in Table 1. Node insertion precision, node deletion recall, and edge deletion recall where consistently at 100% for all models, and edge insertion precision as well as edge deletion precision very close to 100%. This is

the case for both hinge and crossentropy loss. Unsurprisingly, VGAE and VGRNN perform poorly on node edits, for which they are not designed, but also underperform on some edge edit scores; especially VGRNN on edge edit precision. We note that our datasets are strictly Markovian, such that the added recurrent capability of VGRNNs may not provide much value and that, indeed, the difficulty of maintaining the state across node set changes may hurt VGRNN in these cases.

Next, we consider the following two tree dynamical systems, where node labels are represented via one-hot coding.

**Boolean Formulae:** Random Boolean formula with up to three binary operators (e.g. $(x \lor \neg y) \land x \land y$), which get simplified with rules like $x \land \neg x \to \bot$ until the formula could not be simplified anymore.

**Peano addition:** Random additions of single-digit integers with at most $3 +$ operators, which get re-written using Peano's definition of addition, *i.e.* $m + \text{succ}(n) = \text{succ}(m) + n$ and $m + 0 = m$.

The purpose of these datasets is to test whether our GEN is able to correctly predict node attributes. We compare against a Gaussian process prediction approach for tree time series prediction suggested by Paaßen et al. (2018) which uses Gaussian process regression with an RBF kernel on the tree edit distance to predict the next tree in kernel space and then solves a kernel pre-image problem to find the actual next tree (Paaßen et al., 2018). We use the same GEN hyper parameters as for the graph dynamical systems.

As results we observe that the GEN is able to achieve perfect predictive accuracy for both datasets on all five repeats of the experiment. By contrast, the Gaussian process prediction scheme yields an RMSE (as measured by the tree edit distance) of $2.39$ (std.: $0.24$) on Boolean and $4.54$ (std.: $0.77$) on Peano, indicating that the datasets are not trivial to predict.

We evaluated the runtime of GENs on a variation of the HEP-Th paper dataset of Leskovec et al. (2007). In particular, we considered all authors as nodes and included an edge if two authors submitted a joint HEP-Th paper within the last $\tau$ month from some month $t$. We removed authors without any papers from the graph and resolved duplicates. We varied $t$ over the entire range between January 1992 and April 2003 in the HEP-Th dataset and $\tau \in \{1, \ldots, 12\}$, thus obtaining $1554$ different graphs of sizes in the range $[100, 2786]$. For all these graphs, we measured the time needed to compute a forward and backward pass with a GEN without edge filtering (*i.e.* $\vec{e}^+ = \vec{e}^- = \vec{1}$ in all cases) and with edge filtering, respectively. For forward computations, both variants scaled sub-quadratically with empiric exponents of $1.65$ and $1.37$, respectively. For the backward pass, GENs without node filtering scaled with an exponent of $4.11$, whereas GENs with edge filtering scaled roughly linearly (exponent $0.93$), yielding faster times by several orders of magnitude.

We emphasize that edge filtering can not be made arbitrarily strict. As discussed in Section 3.2, the limit on the number of edited nodes must be adjusted to the training data; otherwise accuracy will suffer. Still, the precise value of the limit only influences a constant factor in the linear efficiency.

## 5 CONCLUSION

We introduced the graph edit network, a novel output layer for graph neural networks to predict graph edits. In contrast to prior work, graph edits cover all possible graph-to-graph transformations, including changes in the node set. Importantly, graph edits are sparse, reducing the time complexity from quadratic to linear and facilitating interpretation. Further, graph edits can be locally decided at each node, avoiding the need for pooling layers. In addition to the novel output layer, we also provided a training scheme which only requires a mapping between nodes as input and then returns target values for all outputs of the linear layer, turning the training into combination of binary classification and regression tasks that can be solved with established methods. We further showed that, if the input node mapping is optimal and the graph neural network layers are expressive enough, a graph edit network can achieve edit scripts almost as short as the graph edit distance.

Empirically, we evaluated graph edit networks against variational graph autoencoders, variational graph recurrent nets, and kernel time series prediction on three graph and two tree dynamical systems, proving the concept of graph edit networks. We hope that our work is the starting point for exciting further research in the field of graph-to-graph learning and time series prediction on graphs, especially in combination with recurrent graph neural networks to go beyond the Markovian setting.

ACKNOWLEDGMENTS

Funding by the German Research Foundation (DFG) under grant number PA 3460/1-1 as well as the Bielefeld Young Researchers' Fund is gratefully acknowledged. D.Z. and D.G. gratefully acknowledge the support of the Swiss National Science Foundation under grant 200021/172671.

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

# A  PROOFS

## A.1  PROOF OF THEOREM 1

We structure our proof into three lemmas. Lemma 1 translates a graph mapping into a script, Lemma 2 translates a script to a graph mapping, and Lemma 3 shows that the length never gets longer by doing so. This argument is analogous to Proposition 1 by Bougleux et al. (2017), who show a one-to-one correspondence between bijective mappings and a reduced set of edit scripts, called restricted paths. In this work, we aim to connect the notion of graph mappings to teaching signals for graph edit networks. To make this connection, we require stronger statements about the structure of such graph mappings and a different order of operations compared to Bougleux et al. (2017), such that we provide a full argument here, without relying on Bougleux et al. (2017) directly.

First, we define the corresponding script to a graph mapping.

**Definition 2.** Let $G = (V = \{1, \ldots, M\}, E, \boldsymbol{X})$ and $H = (V' = \{1, \ldots, N\}, E', \boldsymbol{X}')$ be two non-empty graphs, *i.e.* $M, N > 0$, and let $\psi$ be a graph mapping between $G$ and $H$. We then define the script $\bar{\delta}_\psi$ corresponding to $\psi$ between $G$ and $H$ as follows. First, we construct a replacement $\mathrm{rep}_{i,\vec{x}'_{\psi(i)}}$ for all $i$ (in ascending order) where $i \leq M$, $\psi(i) \leq N$ and $\vec{x}_i \neq \vec{x}'_{\psi(i)}$. Let $\bar{\delta}_\psi^{\mathrm{rep}}$ be the script of all these replacements. Next, we construct an insertion $\mathrm{ins}_{x'_{\psi(i)}}$ for all $i$ (in ascending order) where $i > M$ and $\psi(i) \leq N$. Let $\bar{\delta}_\psi^{\mathrm{ins}}$ be the script of all these insertions. Next, we construct an edge insertion $\mathrm{eins}_{i,j}$ for all $(i, j)$ (in lexicographically ascending order) where $(i, j) \notin E$ but $(\psi(i), \psi(j)) \in E'$. Let $\bar{\delta}_\psi^{\mathrm{eins}}$ be the script of all these edge insertions. Next, we construct an edge deletion $\mathrm{edel}_{i,j}$ for all $(i, j)$ (in lexicographically ascending order) where $(i, j) \in E$ but $(\psi(i), \psi(j)) \notin E'$. Let $\bar{\delta}_\psi^{\mathrm{edel}}$ be the script of all these edge deletions. Finally, we construct a node deletion $\mathrm{del}_i$ for all $i \leq M$ (in *descending* order) where $\psi(i) > N$. Let $\bar{\delta}_\psi^{\mathrm{del}}$ be the script of all these node deletions. We then define the overall script $\bar{\delta}_\psi$ as the concatenation $\bar{\delta}_\psi = \bar{\delta}_\psi^{\mathrm{rep}}, \bar{\delta}_\psi^{\mathrm{ins}}, \bar{\delta}_\psi^{\mathrm{eins}}, \bar{\delta}_\psi^{\mathrm{edel}}, \bar{\delta}_\psi^{\mathrm{del}}$.

We next show that the scripts resulting from graph mappings do what they should, *i.e.* they to indeed convert $G$ to a graph that is isomorphic to $H$.

**Lemma 1.** *Let $G = (V = \{1, \ldots, M\}, E, \boldsymbol{X})$ and $H = (V' = \{1, \ldots, N\}, E', \boldsymbol{X}')$ be two non-empty graphs,* i.e. *$M, N > 0$, let $\psi$ be a graph mapping between $G$ and $H$, and let $\bar{\delta}_\psi$ be the corresponding script. Then, $\tilde{H} := \bar{\delta}_\psi(G)$ is isomorphic to $H$.*

*Proof.* In particular, by applying $\bar{\delta}_\psi^{\mathrm{rep}}$ and $\bar{\delta}_\psi^{\mathrm{ins}}$, to $G$, we obtain a node set $\tilde{V}$ and attributes $\tilde{X}$ with the following properties. First, consider $U := \{i \in \tilde{V} | \psi(i) \leq N\}$. Due to our construction and Definition 1 of a graph mapping, it holds $U = \{i \leq M | \psi(i) \leq N\} \cup \{M < i \leq M + |\mathrm{Ins}_\psi|\}$ and $\psi(U) = \{j \leq N | \psi^{-1}(j) \leq M\} \cup \mathrm{Ins}_\psi = V'$. Further, for all $i \in U$ we have by construction $\tilde{x}_i = \vec{x}'_{\psi(i)}$, because either $i \leq M$ and $\vec{x}_i = \vec{x}'_{\psi(i)}$, in which case this holds trivially, or $i \leq M$ and $\vec{x}_i \neq \vec{x}'_{\psi(i)}$, in which case a replacement was applied which ensured the condition, or $M < i \leq M + |\mathrm{Ins}_\psi|$, in which case an insertion was applied that ensured the condition.

Next, we consider the graph $\tilde{G} = (\tilde{V}, \tilde{E}, \tilde{\boldsymbol{X}}) = \bar{\delta}_\psi^{\mathrm{rep}}, \bar{\delta}_\psi^{\mathrm{ins}}, \bar{\delta}_\psi^{\mathrm{eins}}, \bar{\delta}_\psi^{\mathrm{edel}}(G)$, *i.e.* the graph that results from additionally applying all edge insertions and edge deletions. For this graph we can show that the subgraph restricted to the node set $U$ is isomorphic to $H$ with the isomorphism $\psi$ restricted to $U$. In particular, we have already shown that $\psi(U) = V'$ and that $\tilde{x}_i = \vec{x}'_{\psi(i)}$. It remains to show that for all $i, j \in U$ it holds: $(i, j) \in \tilde{E}$ if and only if $(\psi(i), \psi(j)) \in E'$. This, however, is exactly the condition enforced by the edge deletion/insertion construction above. Accordingly, the subgraph restricted to the nodes $U$ is indeed isomorphic to $H$.

Additionally, we also obtain that for any node $i \notin U$, there exists no edge $(i, j)$ or $(j, i)$ in $\tilde{E}$, because these would have been deleted by $\bar{\delta}_\psi^{\mathrm{edel}}$. Finally, $\bar{\delta}_\psi^{\mathrm{del}}$ deletes all nodes $i \notin U$ (in descending order to prevent interference in node ordering) and the remaining graph $\tilde{H} = \bar{\delta}(G)$ is isomorphic to $H$, because it is exactly the subgraph we have analyzed before. This concludes the proof. □

In a next step, we define the conversion back from a script to a graph mapping.

**Definition 3.** Let $G = (V = \{1, \ldots, M\}, E, \boldsymbol{X})$ and $H = (V' = \{1, \ldots, N\}, E', \boldsymbol{X}')$ be two non-empty graphs, *i.e.* $M, N > 0$. Further, let $\tilde{H} = (\tilde{V}' = \{1, \ldots, N\}, \tilde{E}', \tilde{\boldsymbol{X}}')$ be a graph that is isomorphic to $H$ via isomorphism $\tilde{\psi} : \{1, \ldots, N\} \to \{1, \ldots, N\}$ from $\tilde{V}'$ to $V'$, *i.e.* for all $i \in \tilde{V}'$: $\tilde{x}'_i = \vec{x}'_{\psi(i)}$, and $(i, j) \in \tilde{E}'$ if and only if $(\tilde{\psi}(i), \tilde{\psi}(j)) \in E'$. Finally, let $\bar{\delta} = \delta_1, \ldots, \delta_T$ be an edit script, such that $\bar{\delta}(G) = \tilde{H}$. Then, we define the mapping $\psi_{\bar{\delta}} : \{1, \ldots, M+N\} \to \{1, \ldots, M+N\}$ recursively as follows.

First, if $T = 0$ (*i.e.* $\bar{\delta}$ is empty), we set $\psi_{\bar{\delta}} = \tilde{\psi}$, extended by the identity on the inputs $N+1, \ldots, M+N$. Next, if $T > 0$, let $\psi' := \psi_{\delta_2, \ldots, \delta_T}$ and consider the first edit $\delta_1$. If $\delta_1(G) = G$, *i.e.* the edit has no effect, we set $\psi_{\bar{\delta}} = \psi'$.

If $\delta_1 = \text{del}_j$ for some $j$, we set $\psi_{\bar{\delta}}(i) = \psi'(i)$ for all $i < j$, $\psi_{\bar{\delta}}(i) = \psi'(i-1)$ for all $i > j$, and $\psi_{\bar{\delta}}(i) = M + N$.

If $\delta_1 = \text{ins}_{\vec{x}}$ for some $\vec{x}$, let $i^* := \psi'^{-1}(M+N+1)$ and distinguish four cases. First, if $i^* \leq M$ and $\psi'(M+1) \leq N$, we set $\psi_{\bar{\delta}}(i) = \psi'(i)$ for all $i \neq i^*$ and $\psi_{\bar{\delta}}(i^*) = \psi'(M+N+1)$. Second, if $i^* \leq M$ and $\psi'(M+1) > N$, we set $\psi_{\bar{\delta}}(i) = \psi'(i)$ for all $i \neq i^*$ with $i \leq M$, $\psi_{\bar{\delta}}(i^*) = \psi'(M+1)$, and $\psi_{\bar{\delta}}(i) = \psi'(i+1)$ for all $i > M$. Third, if $i^* = M+1$, we set $\psi_{\bar{\delta}}(i) = \psi'(i)$ for all $i \leq M$ and $\psi_{\bar{\delta}}(i) = \psi'(i+1)$ for all $i > M$. Fourth, if $i^* > M+1$, we set $\psi_{\bar{\delta}}(i) = \psi'$, restricted to $\{1, \ldots, M+N\}$.

Finally, if $\delta_1$ is any other edit, we set $\psi_{\bar{\delta}} = \psi'$.

We will later prove in Corollary 2 that the resulting mapping $\psi_{\bar{\delta}}$ is indeed a graph mapping.

Refer to Figure 3 for an example of the construction. In particular, we consider the script $\bar{\delta} = \text{del}_2, \text{ins}_{\vec{x}_1}, \text{ins}_{\vec{x}_2}, \text{del}_3$, transforming the graph $G = (V = \{1, 2, 3\}, E = \emptyset, \boldsymbol{X})$ to the graph $\tilde{H} = (\tilde{V}' = \{1, 2, 3\}, \tilde{E}' = \emptyset, \tilde{\boldsymbol{X}}')$, which in turn is isomorphic to the graph $H = (V' = \{1, 2, 3\}, E' = \emptyset, \boldsymbol{X}')$ for some attributes $\boldsymbol{X}, \tilde{\boldsymbol{X}}', \boldsymbol{X}'$, and for the isomorphism $\tilde{\psi}$ between $\tilde{H}$ and $H$ with $\tilde{\psi}(1) = 3$, $\tilde{\psi}(2) = 1$, and $\tilde{\psi}(3) = 2$.

Now, we construct the mapping $\psi_{\bar{\delta}}$ recursively from the last edit to the first. The initial mapping for the empty script is $\psi_\epsilon : \{1, \ldots, 3+3\} \to \{1, \ldots, 3+3\}$ with $\psi_\epsilon(i) = \tilde{\psi}(i)$ for $i \leq 3$ and $\psi_\epsilon(i) = i$ for $i > 3$. Next, we incorporate the last edit of the script $\text{del}_3$. Because this is a deletion, the graph before the deletion must have had one node more. Accordingly, $\psi_{\text{del}_3}$ is now defined on the domain and image $\{1, \ldots, 4+3\}$. In more detail, following Definition 3 we obtain the mapping $\psi_{\text{del}_3}(1) = \psi_\epsilon(1) = 3$, $\psi_{\text{del}_3}(2) = \psi_\epsilon(2) = 1$, $\psi_{\text{del}_3}(3) = 4+3 = 7$, $\psi_{\text{del}_3}(4) = \psi_\epsilon(4-1) = 2$, $\psi_{\text{del}_3}(5) = \psi_\epsilon(5-1) = 4$, $\psi_{\text{del}_3}(6) = \psi_\epsilon(6-1) = 5$, and $\psi_{\text{del}_3}(7) = \psi_\epsilon(7-1) = 6$. In the figure, this mapping is obtained by following all arrows from the graph right of $\tilde{H}$ to their end point.

Next, we incorporate the edit $\text{ins}_{\vec{x}_2}$. Because we consider an insertion, the graph before the insertion had one node less. Accordingly, the mapping $\psi_{\text{ins}_{\vec{x}_2}, \text{del}_3}$ is now defined on the domain and image $\{1, \ldots, 3+3\}$; it is crucial that we remove the entry $3+3+1 = 7$ from the codomain of our mapping. The pre-image of 7 is $i^* = \psi_{\text{del}_3}^{-1}(7) = 3$. Further, we need to consider whether the inserted node $3+1 = 4$ gets deleted later in the script. Because $\psi_{\text{del}_3}(3+1) = 2 \leq 3$, we know that this is not the case. Accordingly, the first of the four cases in Definition 3 applies and we obtain $\psi_{\text{ins}_{\vec{x}_2}, \text{del}_3}(i) = \psi_{\text{del}_3}(i)$ for all $i \neq i^* = 3$ and $\psi_{\text{ins}_{\vec{x}_2}, \text{del}_3}(3) = \psi_{\text{del}3}(7) = 6$. Note that this is still a proper graph mapping, *i.e.* the bijectivity is never violated and the only node $j \in \text{Ins}_{\psi_{\text{ins}_{\vec{x}_2}, \text{del}_3}}$ is 2 with the pre-image $\psi_{\text{ins}_{\vec{x}_2}, \text{del}_3}(2) = 4 = 3+1$, as required.

Now, we incorporate another insertion $\text{ins}_{\vec{x}_1}$. We observe that the newly inserted node $2+1 = 3$ will get deleted later, since we have $\psi_{\text{ins}_{\vec{x}_2}, \text{del}_3}(2+1) = 6 > 3$. Furthermore, the newly inserted node 3 is at the same time the pre-image of entry $2+3+1 = 6$, which we need to delete later. Accordingly, the third case in Definition 3 applies and we obtain $\psi_{\text{ins}_{\vec{x}_1}, \text{ins}_{\vec{x}_2}, \text{del}_3}(i) = \psi_{\text{ins}_{\vec{x}_2}, \text{del}_3}(i)$ for $i \leq 2$ and $\psi_{\text{ins}_{\vec{x}_1}, \text{ins}_{\vec{x}_2}, \text{del}_3}(i) = \psi_{\text{ins}_{\vec{x}_2}, \text{del}_3}(i+1)$ for $i > 2$. Because we do not refer to $\psi_{\text{ins}_{\vec{x}_2}, \text{del}_3}(2+1) = 6$, we do not leave the desired range $\{1, \ldots, 2+3\}$. Also note that $\text{Ins}_{\psi_{\text{ins}_{\vec{x}_1}, \text{ins}_{\vec{x}_2}, \text{del}_3}}$ still contains only the node 2, which has the pre-image $\psi_{\text{ins}_{\vec{x}_1}, \text{ins}_{\vec{x}_2}, \text{del}_3}^{-1}(2) = 3 = 2+1$ as required.

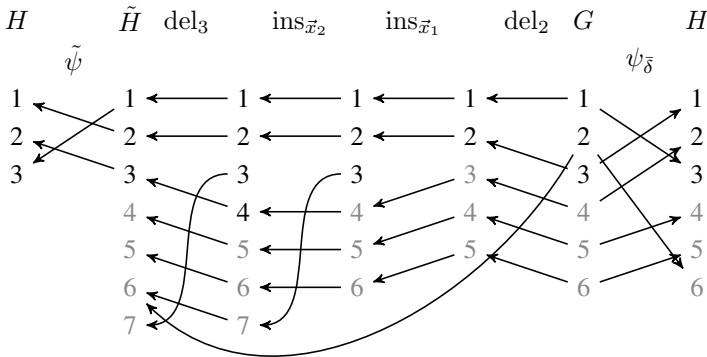

Figure 3: An illustration of the conversion of a script $\bar{\delta} = \text{del}_2, \text{ins}_{\vec{x}_1}, \text{ins}_{\vec{x}_2}, \text{del}_3$ to a graph mapping $\psi_{\bar{\delta}}$. We start with an initial isomorphism $\tilde{\psi}$ between the graph $H$ and the graph $\tilde{H} = \bar{\delta}(G)$ (left) and then adjust the mapping from left to right until we arrive at a final graph map between $G$ and $H$ (right). The mapping is obtained by following the arrows from $G$ to $H$. For simplicity we assume that the graphs have no edges. Also note that we include the 'virtual nodes' $M + 1, \ldots, M + N$ in the illustration.

Finally, we incorporate the deletion $\text{del}_2$. As stated in Definition 3, we thus obtain $\psi_{\bar{\delta}}(i) = \psi_{\text{ins}_{\vec{x}_1}, \text{ins}_{\vec{x}_2}, \text{del}_3}(i)$ for $i < 2$, $\psi_{\bar{\delta}}(2) = 3 + 3 = 6$, and $\psi_{\bar{\delta}}(i) = \psi_{\text{ins}_{\vec{x}_1}, \text{ins}_{\vec{x}_2}, \text{del}_3}(i - 1)$ for $i > 2$. The final mapping is thus $\psi_{\bar{\delta}}(1) = 3$, $\psi_{\bar{\delta}}(2) = 6$, $\psi_{\bar{\delta}}(3) = 1$, $\psi_{\bar{\delta}}(4) = 2$, $\psi_{\bar{\delta}}(5) = 4$, and $\psi_{\bar{\delta}}(6) = 5$. In Figure 3, we obtain this mapping by starting at $G$ and following the arrows to their end points.

Our next step is to show that the mapping constructed from a script is indeed a graph mapping and that additional structure applies, which we can exploit later on.

**Lemma 2.** *Let $G, \tilde{H}, H$ be graphs as in the previous definition with isomorphism $\tilde{\psi}$ between $\tilde{H}$ and $H$, let $\bar{\delta} = \delta_1, \ldots, \delta_T$ be a script, such that $\bar{\delta}(G) = \tilde{H}$, let $\psi_{\bar{\delta}}$ be the corresponding mapping, and let $\psi : \{1, \ldots, M + N\} \to \{1, \ldots, M + N\}$ be defined as $\psi(i) = \tilde{\psi}^{-1}(\psi_{\bar{\delta}}(i))$ if $\psi_{\bar{\delta}}(i) \leq N$ and $\psi(i) = \psi_{\bar{\delta}}(i)$ otherwise, i.e. the mapping composed with $\tilde{\psi}^{-1}$. Further, let $\text{Del}_\psi := \{i \leq M | \psi(i) > N\}$. Then, $\psi$ is bijective. Further, for all $i \notin \text{Del}_\psi$, $\psi$ is monotonously increasing (constraint 1), and the image of $\psi$ on $\text{Del}_\psi$ is $\{M + N + 1 - |\text{Del}_\psi|, \ldots, M + N\}$ (constraint 2).*

*Proof.* We perform this proof via induction over the length $T$ of the script $\bar{\delta} = \delta_1, \ldots, \delta_T$. If $T = 0$, $\psi$ is the identity, which is obviously bijective, yields $\text{Del}_\psi = \emptyset$ (such that constraint 2 is fulfilled), and is monotonously increasing (such that constraint 1 is fulfilled).

Now, if $T > 1$, let $\psi'$ be defined as $\psi$ above, but between the graphs $\delta_1(G)$ and $H$. Accordingly, $\psi'$ is a bjective mapping from and to $\{1, \ldots, M' + N\}$ by induction, where $M'$ is the size of the node set of $\delta_1(G)$. Further, by induction we also know that $\psi'$ is monotonously increasing on all inputs except on $\text{Del}_{\psi'} := \{i \leq M' | \psi'(i) > N\}$ (constraint 1) and that the image of $\psi'$ on $\text{Del}_{\psi'}$ is precisely $\{M' + N + 1 - |\text{Del}_{\psi'}|, \ldots, M' + N\}$ (constraint 2). Now, consider the first edit $\delta_1$. If $\delta_1$ leaves the node set as-is, we obtain $M = M'$, $\psi = \psi'$, and both constraints hold by induction.

If $\delta_1 = \text{del}_j$ for some $j \leq M$, we obtain $M' = M - 1$. Further, by the definition of $\psi_{\bar{\delta}}$, we obtain $\psi(i) = \psi'(i)$ for $i < j$, $\psi(i) = \psi'(i - 1)$ for $i > j$, and $\psi(j) = M + N$. Accordingly, $\psi$ maps the set $\{1, \ldots, j - 1\} \cup \{j + 1, \ldots, M + N\}$ bijectively to $\{1, \ldots, M + N - 1\}$ and the setting $\psi(j) = M + N$ completes the bijective map. Further, $j \in \text{Del}_\psi$ and is mapped to $M + N$, conforming to constraint 2. Finally, for all other inputs both constraints hold by induction.

If $\delta_1 = \text{ins}_{\vec{x}}$ for some $\vec{x}$, we obtain $M' = M + 1$ and thus $\psi'$ is a bijective map from and to $\{1, \ldots, M + N + 1\}$. Now, let $i^* = \psi'^{-1}(M + N + 1)$ and distinguish four cases.

First, if $i^* \leq M$ and $\psi'(M+1) \leq N$ we obtain $\psi(i) = \psi'(i)$ for all $i \neq i^*$ and $\psi(i^*) = \psi'(M+N+1)$. First, observe that $\psi'(M+N+1) < M+N+1$, otherwise $\psi'(M+N+1) = M+N+1$ and $i^* = M+N+1 > M$, which is a contradiction. Accordingly, $\psi$ is a bijective map from and to $\{1, \ldots, M+N\}$ because by definition $\psi$ maps the set $\{1, \ldots, i^*-1\} \cup \{i^*+1, \ldots, M+N\}$ to $\{1, \ldots, M+N\} \setminus \{\psi'(M+N+1)\}$ and the setting $\psi(i^*) = \psi'(M+N+1)$ completes the bijective map. Furthermore, we obtain $\psi'(M+N+1) > N$. Otherwise, the monotonicity of $\psi'$ would require all values $M+2, \ldots, M+N$ to be mapped to values $\leq N$ as well. However, there exist only $N$ such values and one is already blocked by $\psi'(M+1)$, which is a contradiction. Accordingly, $\psi'(M+N+1) > N$ and thus $i^* \in \mathrm{Del}_\psi$. Both constraints hold by induction.

Second, if $i^* \leq M$ and $\psi'(M+1) > N$ we obtain $\psi_{\bar\delta}(i) = \psi'(i)$ for all $i \neq i^*$ with $i \leq M$, $\psi_{\bar\delta}(i^*) = \psi'(M+1)$, and $\psi_{\bar\delta}(i) = \psi'(i+1)$ for all $i > M$. Accordingly, $\psi$ is a bijective map from and to $\{1, \ldots, M+N\}$ because by definition $\psi$ maps the set $\{1, \ldots, i^*-1, i^*+1 \ldots, M+N\}$ to the image of $\psi'$ for the inputs $\{1, \ldots, i^*-1, i^*+1 \ldots, M, M+2, \ldots, M+N+1\}$, which is exactly $\{1, \ldots, M+N\} \setminus \{\psi'(M+1)\}$. Setting $\psi_{\bar\delta}(i^*) = \psi'(M+1)$ then completes the bijective map. Further, on all inputs except $i^*$ both constraints still hold due to induction. For $i^*$, we obtain $i^* \in \mathrm{Del}_\psi$ because $\psi(i^*) = \psi'(M+1) > N$ and because $M+1 \in \mathrm{Del}_{\psi'}$ constraint 2 also holds for $i^*$ due to induction.

Third, if $i^* = M+1$, we obtain $\psi_{\bar\delta}(i) = \psi'(i)$ for all $i \leq M$, $\psi_{\bar\delta}(i) = \psi'(i+1)$ for all $i > M$. This is a bijective map from and to $\{1, \ldots, M+N\}$ because by definition $\psi$ maps $\{i, \ldots, M+N\}$ to the image of $\psi'$ for $\{1, \ldots, M, M+2, \ldots, M+N+1\}$, which is exactly $\{1, \ldots, M+N\}$. Further, the constraints hold due to induction.

Fourth, if $i^* > M+1$, we obtain $\psi_{\bar\delta}(i) = \psi'$ restricted to $\{1, \ldots, M+N\}$. In this case, we observe that $i^*$ must be $M+N+1$. Otherwise, we would obtain $\psi'(i^*+1) < M+N+1 = \psi'(i^*)$, which contradicts monotonicity of $\psi'$. Accordingly, restricting $\psi'$ to $\{1, \ldots, M+N\}$ does indeed yield a bijective map to $\{1, \ldots, M+N\}$ which conforms to both constraints due to induction.

Because this covers all possible edits, this concludes our proof. $\qquad\square$

**Corollary 2.** *Let $G$, $H$, $\tilde{H}$, $\tilde{\psi}$, $\bar\delta$, $\psi_{\bar\delta}$, and $\psi$ be defined as in the previous Lemma. then, $\psi_{\bar\delta}$ is a graph mapping between $G$ and $H$.*

*Proof.* First, observe that $\psi_{\bar\delta}(i) = \tilde\psi(\psi(i))$ for all $i$ with $\psi_{\bar\delta}(i) \leq N$, and $\psi_{\bar\delta}(i) = \psi(i)$ otherwise. Further, observe that $\psi_{\bar\delta}$ must thus be bijective. Otherwise, $\psi$ or $\tilde\psi$ could not be bijective. Next, assume that there exists some $j \in \mathrm{Ins}_{\psi_{\bar\delta}}$ such that $\psi_{\bar\delta}^{-1}(j) > M + |\mathrm{Ins}_{\psi_{\bar\delta}}|$. Then, because $\psi_{\bar\delta}$ is bijective, there also exists some $i$ with $M < i \leq M + |\mathrm{Ins}_{\psi_{\bar\delta}}|$ with $\psi_{\bar\delta}(i) \notin \mathrm{Ins}_{\psi_{\bar\delta}}$. This implies that $\psi_{\bar\delta}(i) > N \geq j$, even though $\psi_{\bar\delta}^{-1}(j) > i$, which contradicts the monotonicity constraint on $\psi$. Accordingly, $\psi_{\bar\delta}$ is indeed a graph mapping between $G$ and $H$. $\qquad\square$

**Lemma 3.** *Let $G = (V = \{1, \ldots, M\}, E, \boldsymbol{X})$ and $H = (V' = \{1, \ldots, N\}, E', \boldsymbol{X'})$ be two non-empty graphs, i.e. $M, N > 0$. Further, let $\tilde{H} = (\tilde{V}' = \{1, \ldots, N\}, \tilde{E}', \tilde{\boldsymbol{X}}')$ be a graph that is isomorphic to $H$ via isomorphism $\tilde\psi : \{1, \ldots, N\} \to \{1, \ldots, N\}$, and let $\bar\delta$ be an edit script with $\bar\delta(G) = \tilde{H}$. Finally, let $\psi_{\bar\delta}$ be the corresponding graph mapping between $G$ and $H$ and let $\bar\delta_{\psi_{\bar\delta}}$ be the edit script corresponding to that graph mapping. Then, it holds: $|\bar\delta_{\psi_{\bar\delta}}| \leq |\bar\delta|$.*

*Proof.* We again prove this claim via induction over the length $T$ of the script $\bar\delta = \delta_1, \ldots, \delta_T$.

First, if $T = 0$, $G = \bar\delta(G) = \tilde{H}$. Accordingly, $\psi_{\bar\delta}$ is $\tilde\psi$ (extended with the identity on $M+1, \ldots, M+N$). It then follows that $\bar\delta_{\psi_{\bar\delta}}$ is empty. In particular, there can not exist any $i \leq M$ with $\vec{x}_i \neq \vec{x}'_{\psi_{\bar\delta}(i)} = \vec{x}'_{\tilde\psi(i)}$, otherwise $\tilde\psi$ would not be an isomorphism between $\tilde{H}$ and $H$. Accordingly, $\bar\delta^{\mathrm{rep}}_{\psi_{\bar\delta}}$ is empty. Further, there can not exist any $i > M$ with $\psi_{\bar\delta}(i) \leq N$, because $\psi_{\bar\delta}(i) = i > M = N$. Accordingly, $\bar\delta^{\mathrm{ins}}_{\psi_{\bar\delta}}$ is empty. Next, for all $i, j \leq M$ it holds $(i, j) \in E \iff (\psi_{\bar\delta}(i), \psi_{\bar\delta}(j)) = (\tilde\psi(i), \tilde\psi(j)) \in E'$, otherwise $\tilde\psi$ would not be an isomorphism between $\tilde{H}$ and $H$; and for all $i, j > M = N$ it holds $(i, j) \notin E$ and $(\psi_{\bar\delta}(i), \psi_{\bar\delta}(j)) = (i, j) \notin E'$ per definition. Accordingly, both $\bar\delta^{\mathrm{eins}}_{\psi_{\bar\delta}}$ and $\bar\delta^{\mathrm{edel}}_{\psi_{\bar\delta}}$ are empty. Finally, there can not exist an $i \leq M$ with $\psi_{\bar\delta}(i) = \tilde\psi(i) > N$ because the image of $\tilde\psi$ is

$\{1, \ldots, N\}$. Accordingly, $\bar{\delta}_{\psi_{\bar{\delta}}}^{\mathrm{del}}$ is empty and thus $\bar{\delta}_{\psi_{\bar{\delta}}} = \bar{\delta}_{\psi_{\bar{\delta}}}^{\mathrm{rep}} \bar{\delta}_{\psi_{\bar{\delta}}}^{\mathrm{ins}} \bar{\delta}_{\psi_{\bar{\delta}}}^{\mathrm{eins}} \bar{\delta}_{\psi_{\bar{\delta}}}^{\mathrm{edel}} \bar{\delta}_{\psi_{\bar{\delta}}}^{\mathrm{del}}$ is overall empty as well, *i.e.* $|\bar{\delta}_{\psi_{\bar{\delta}}}| = 0 \leq 0 = |\bar{\delta}|$, as claimed.

Now, consider the case $T > 0$, let $\psi' := \psi_{\delta_2, \ldots, \delta_T}$, and let $\tilde{G} = (\tilde{V}, \tilde{E}, \tilde{\boldsymbol{X}}) := \delta_1(G)$. Further, let $\bar{\delta}_{\psi'}$ be the script that $\psi'$ is transformed into based on $\tilde{G}$ and $H$. By induction, $|\bar{\delta}_{\psi'}| \leq T - 1$. We now consider the first edit $\delta_1$.

If $\delta_1$ has no effect on $G$, we obtain $\psi_{\bar{\delta}} = \psi'$ and by definition $\bar{\delta}_{\psi_{\bar{\delta}}} = \bar{\delta}_{\psi'}$, which in turn yields $|\bar{\delta}_{\psi_{\bar{\delta}}}| = |\bar{\delta}_{\psi'}| \leq T - 1 < T$ as claimed.

If $\delta_1 = \mathrm{rep}_{i,\vec{x}}$ for some $i \leq M$ and some $\vec{x}$, we obtain $\psi_{\bar{\delta}} = \psi'$, $V = \tilde{V}$, $E = \tilde{E}$, $\vec{x}_j = \tilde{x}_j$ for $j \neq i$, and $\tilde{x}_i = \vec{x} \neq \vec{x}_i$. Now, distinguish two cases. If $\vec{x} = \vec{x}'_{\psi'(i)}$, *i.e.* $\tilde{x}_i = \vec{x}'_{\psi'(i)} \neq \vec{x}_i$, $\bar{\delta}_{\psi_{\bar{\delta}}}$ contains by construction an edit $\mathrm{rep}_{i,\vec{x}}$ and otherwise the same edits as $\bar{\delta}_{\psi'}$, such that we obtain $|\bar{\delta}_{\psi_{\bar{\delta}}}| = |\bar{\delta}_{\psi'}| + 1 \leq T - 1 + 1 = T$ as claimed. If $\vec{x} \neq \vec{x}'_{\psi'(i)}$, *i.e.* $\delta_1$ sets $\tilde{x}_i$ to a value which is temporary and not equal to the final value of $\vec{x}'_{\psi(i)}$, we obtain $\bar{\delta}_{\psi_{\bar{\delta}}} = \bar{\delta}_{\psi'}$, because $\bar{\delta}_{\psi'}$ by construction already contains an edit $\mathrm{rep}_{i,\vec{x}'_{\psi'(i)}}$. Accordingly, we obtain $|\bar{\delta}_{\psi_{\bar{\delta}}}| = |\bar{\delta}_{\psi'}| \leq T - 1 < T$ as claimed.

If $\delta_1 = \mathrm{ins}_{\vec{x}}$ for some $\vec{x}$, we obtain $\tilde{V} = V \cup \{M + 1\}$, $\tilde{E} = E$, $\tilde{x}_i = \vec{x}_i$ for $i \leq M$, and $\tilde{x}_{M+1} = \vec{x}$. Now, let let $i^* := {\psi'}^{-1}(M + N + 1)$ and distinguish four cases.

First, if $i^* \leq M$ and $\psi'(M + 1) \leq N$, we obtain $\psi_{\bar{\delta}}(i) = \psi'(i)$ for $i \neq i^*$ and $\psi_{\bar{\delta}}(i^*) = \psi'(M + N + 1)$. As argued in the proof of the previous lemma, $\psi'(M + N + 1) > N$ due to monotonicity of $\psi'$. Accordingly, $i^*$ is deleted both in $\bar{\delta}_{\psi'}$ and in $\bar{\delta}_{\psi_{\bar{\delta}}}$. Further, $M + 1$ is inserted in $\bar{\delta}_{\psi_{\bar{\delta}}}$ but not in $\bar{\delta}_{\psi'}$, while all other edits remain the same such that we obtain $|\bar{\delta}_{\psi_{\bar{\delta}}}| = |\bar{\delta}_{\psi'}| + 1 \leq T - 1 + 1 = T$ as claimed.

Second, if $i^* \leq M$ and $\psi'(M + 1) > N$, we obtain $\psi_{\bar{\delta}}(i) = \psi'(i)$ for all $i$ with $i \leq M$ and $i \neq i^*$, $\psi_{\bar{\delta}}(i^*) = \psi'(M + 1)$, and $\psi_{\bar{\delta}}(i) = \psi'(i + 1)$ for all $i > M$. Because $\psi'(M + 1) > N$, $i^*$ is deleted both in $\bar{\delta}_{\psi'}$ and in $\bar{\delta}_{\psi_{\bar{\delta}}}$. Further, $M + 1$ is deleted in $\bar{\delta}_{\psi'}$ but not in $\bar{\delta}_{\psi_{\bar{\delta}}}$, while all other edits remain the same, such that we obtain $|\bar{\delta}_{\psi_{\bar{\delta}}}| = |\bar{\delta}_{\psi'}| - 1 \leq T - 2 < T$ as claimed.

Third, if $i^* = M + 1$, we obtain $\psi_{\bar{\delta}}(i) = \psi'(i)$ for all $i \leq M$ and $\psi_{\bar{\delta}}(i) = \psi'(i + 1)$ otherwise. Again, $M + 1$ is deleted in $\bar{\delta}_{\psi'}$ but not in $\bar{\delta}_{\psi_{\bar{\delta}}}$, while all other edits remain the same, such that we obtain $|\bar{\delta}_{\psi_{\bar{\delta}}}| = |\bar{\delta}_{\psi'}| - 1 \leq T - 2 < T$ as claimed.

Fourth, if $i^* > M + 1$, we obtain $\psi_{\bar{\delta}} = \psi'$ restricted to $\{1, \ldots, M + N\}$. Further, $i^* = M + N + 1$ as argued in the proof of the previous lemma. Accordingly, $M + 1$ is inserted in $\bar{\delta}_{\psi_{\bar{\delta}}}$ but not in $\bar{\delta}_{\psi'}$, while all other edits remain the same, such that we obtain $|\bar{\delta}_{\psi_{\bar{\delta}}}| = |\bar{\delta}_{\psi'}| + 1 \leq T - 1 + 1 = T$ as claimed.

Next, if $\delta_1 = \mathrm{eins}_{i,j}$, we obtain $\psi_{\bar{\delta}} = \psi'$ as well as $\tilde{V} = V$, $\tilde{X} = X'$, and $\tilde{E} = E \cup \{(i, j)\}$. Now, distinguish two cases. If $(\psi'(i), \psi'(j)) \in E'$, $\bar{\delta}_{\psi_{\bar{\delta}}}$ inserts $(i, j)$ whereas $\bar{\delta}_{\psi'}$ does not, while all other edits remain the same, such that we obtain $|\bar{\delta}_{\psi_{\bar{\delta}}}| = |\bar{\delta}_{\psi'}| + 1 \leq T - 1 + 1 = T$ as claimed. Conversely, if $(\psi'(i), \psi'(j)) \notin E'$, $\bar{\delta}_{\psi'}$ deletes $(i, j)$ whereas $\bar{\delta}_{\psi_{\bar{\delta}}}$ does not, while all other edits remain the same, such that we obtain $|\bar{\delta}_{\psi_{\bar{\delta}}}| = |\bar{\delta}_{\psi'}| - 1 \leq T - 2 < T$ as claimed.

Next, if $\delta_1 = \mathrm{edel}_{i,j}$, we obtain $\psi_{\bar{\delta}} = \psi'$ as well as $\tilde{V} = V$, $\tilde{X} = X'$, and $\tilde{E} = E \setminus \{(i, j)\}$. Now, distinguish two cases. If $(\psi'(i), \psi'(j)) \notin E'$, $\bar{\delta}_{\psi_{\bar{\delta}}}$ deletes $(i, j)$ whereas $\bar{\delta}_{\psi'}$ does not, while all other edits remain the same, such that we obtain $|\bar{\delta}_{\psi_{\bar{\delta}}}| = |\bar{\delta}_{\psi'}| + 1 \leq T - 1 + 1 = T$ as claimed. Conversely, if $(\psi'(i), \psi'(j)) \in E'$, $\bar{\delta}_{\psi'}$ inserts $(i, j)$ whereas $\bar{\delta}_{\psi_{\bar{\delta}}}$ does not, while all other edits remain the same, such that we obtain $|\bar{\delta}_{\psi_{\bar{\delta}}}| = |\bar{\delta}_{\psi'}| - 1 \leq T - 2 < T$ as claimed.

Finally, if $\delta_1 = \mathrm{del}_i$, we obtain $\psi_{\bar{\delta}}(j) = \psi'(j)$ for all $j < i$, $\psi_{\bar{\delta}}(j) = \psi'(j - 1)$ for all $j > i$, and $\psi_{\bar{\delta}}(j) = M + N$, as well as $\tilde{V} = \{1, \ldots, M - 1\}$, $\tilde{x}_j = \vec{x}_j$ for all $j \in \tilde{V}$, and $\tilde{E} = E$. Accordingly, $\bar{\delta}_{\psi_{\bar{\delta}}}$ deletes $i$ whereas $\bar{\delta}_{\psi'}$ does not, while all other edits remain the same, such that we obtain $|\bar{\delta}_{\psi_{\bar{\delta}}}| = |\bar{\delta}_{\psi'}| + 1 \leq T - 1 + 1 = T$ as claimed.

Because this covers all possible edits, this concludes our proof. $\qquad\square$

We note again that the entire argument up to this point is analogous to Proposition 1 of Bougleux et al. (2017). However, in contrast to Bougleux et al. (2017), we do not rely on a notion of restricted edit scripts, instead comparing to all possible edit scripts, and we provide stronger results regarding the structure of our graph mappings via Lemma 2. We also slightly change the order of operations for consistency with Algorithm 1.

### A.2 PROOF OF THEOREM 2

The heart of our proof is Algorithm 2, which translates a graph mapping to a corresponding teaching signal for a graph edit network. We prove the correctness of this algorithm with the following Lemma. Refer to Figure 4 for a graphical intuition.

In particular we consider in this example the graphs $G = (\{1, 2, 3\}, \{(1, 2), (1, 3), (3, 1)\}, \boldsymbol{X})$ and $H = (\{1, 2, 3, 4, 5, 6\}, \{(1, 2), (2, 3), (3, 1), (4, 5), (5, 4)\}, \boldsymbol{X}')$ as well as the graph mapping $\psi$ with $\psi(1) = 3$, $\psi(2) = 2$, and $\psi(3) = 1$. Note that $H$ is not fully connected. In particular, we have the connected components $\{1, 2, 3\}$, $\{4, 5\}$, and $\{6\}$. In itself, this would not be a problem, but note that Algorithm 1 can only insert a new node that is connected to an existing node. Accordingly, we can not insert node without any connection to the pre-existing graph. However, this would be required for the insertion of nodes $4$, $5$, and $6$. To manage this, Algorithm 2 first constructs auxiliary edges (in lines 2-5) connecting the new nodes to the already existing part of the graph. In our example, these auxiliary edges are $(3, 4)$ and $(3, 6)$. Then, the algorithm computes (in lines 6-9) the shortest path to any inserted node from any already existing node, yielding the paths $\pi^4 = (3, 4)$, $\pi^5 = (3, 4, 5)$, and $\pi^6 = (3, 6)$. Line 13 then re-defines the shortest paths as a shortest-path tree with children $\mathrm{ch}(3) = (4, 6)$ and $\mathrm{ch}(4) = (5)$.

Lines 14-22 then take care of performing the actual node insertions, where insertions are put into different steps of the teaching signal if they can not be performed at the same time. In particular, we need to use multiple steps whenever a single node has multiple children in the shortest-path tree and whenever a node that does not yet exist needs to make an insertion. Both is covered by the depth-first-search in lines 14-22. In particular, we first insert parents, then children, and we insert children in succession. In our example, this results in node $4$ being inserted in the first step (by node $1$) and nodes $5$ and $6$ being inserted in the second step (by nodes $1$ and $4$, respectively).

Then, line 24 initializes the third and final step of our teaching signal which performs all remaining edits. In particular, line 25 sets up all replacements (irrelevant in this case), line 26 all edge insertions $((2, 1), (3, 2),$ and $(5, 4)$ in this case), line 27 all edge deletions $((1, 2), (3, 1), (1, 4),$ and $(1, 6)$ in this case), and line 28 all node deletions (none in this case). Lines 29-30 ensure that the edge filter scores $\vec{e}_{K+1}^+$ and $\vec{e}_{K+1}^-$ are consistent with the edge edit scores $\boldsymbol{\mathcal{E}}_{K+1}$. Finally, we return the result.

The resulting script is shown in the bottom of the figure. Note that this script is not as short as possible because it contains four additional auxiliary edits, namely the edge insertions $(1, 4)$ and $(1, 6)$, which are then deleted in the end. These additional edges occur because we need to insert new nodes connected to existing nodes. Accordingly, the disconnected components $\mathcal{C}_1$ and $\mathcal{C}_2$ from $H$ are first connected and then disconnected at the end of the script, yielding the desired graph.

We first consider the time complexity of the algorithm. First, note that computing connected components in line 2 is possible in $\mathcal{O}(N)$. Similarly, the minimum computations in lines 3-4 are possible in $\mathcal{O}(N)$. Constructing the auxiliary graph in line 5 may need $\mathcal{O}(N^2)$ for a dense graph. Because we have unit edge weights for the shortest path computation, a simple breadth-first-search suffices, making the computation in line 7 $\mathcal{O}(N)$. Because we need to repeat this computation for each insertion, lines 6-9 are overall in $\mathcal{O}(N^2)$.

Lines 10-12 are slightly implementation dependent. In principle, constructing $\boldsymbol{\mathcal{E}}_k$ for each step would require $\mathcal{O}((M + N)^2)$. But because this matrix is always zero, we can construct it once and re-use it for every step. All other operations are linear, yielding $\mathcal{O}((M + N)^2)$ overall.

The computation of the children in the shortest-path tree in lines 13 is linear in the number of insertions, which is in $\mathcal{O}(N)$.

Lines 14-22 perform a depth first search through the shortest-path trees, which is in $\mathcal{O}(N)$ again. This also includes recording the inserted edges for line 23.

shortest paths (lines 6-9):
$\pi^4 = (3,4)$, $\pi^5 = (3,4,5)$, $\pi^6 = (3,6)$

Shortest path tree (line 13):
$\mathrm{ch}(3) = (4,6), \mathrm{ch}(4) = (5)$

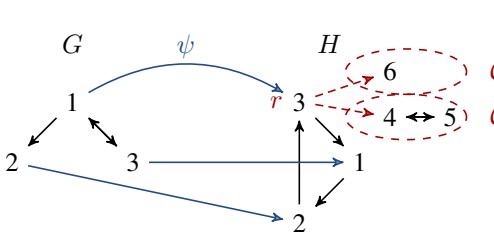

insertions (lines 14-22):
$\nu_{1,1} = +1, \nu_{2,1} = +1, \nu_{2,4} = +1$

edge insertions (line 26):
$\epsilon_{3,2,1} = \epsilon_{3,3,2} = \epsilon_{3,5,4} = +1$

edge deletions (line 27):
$\epsilon_{3,1,2} = \epsilon_{3,3,1} = \epsilon_{3,1,4} = \epsilon_{3,1,6} = -1$

script: $\bar{\delta} = \mathrm{ins}, \mathrm{eins}_{1,4}, \mathrm{ins}, \mathrm{eins}_{4,5}, \mathrm{ins}, \mathrm{eins}_{1,6}, \mathrm{eins}_{2,1}, \mathrm{eins}_{3,2}, \mathrm{eins}_{5,4},$
$\mathrm{edel}_{1,2}, \mathrm{edel}_{1,4}, \mathrm{edel}_{1,6}, \mathrm{edel}_{3,1}$

Figure 4: A graphical illustration of the construction of a teaching signal via Algorithm 2 for the inputs $G = (\{1,2,3\}, \{(1,2),(1,3),(3,1)\}, \boldsymbol{X})$, $H = (\{1,2,3,4,5,6\}, \{(1,2),(2,3),(3,1),(4,5),(5,4)\}, \boldsymbol{X'})$, and $\psi$ with $\psi(1) = 3$, $\psi(2) = 2$, and $\psi(3) = 1$. The mapping $\psi$ is shown in blue, the auxiliary construction of $\tilde{H}$ (lines 2-5 in the algorithm) in red and dashed. After constructing $\tilde{H}$, we first compute shortest paths from 3 to each node in $\mathcal{C}_1$ and $\mathcal{C}_2$ (top right). This defines the shortest path tree with children $\mathrm{ch}(3) = (4,6)$ and $\mathrm{ch}(4) = 5$. Accordingly, node $\psi^{-1}(3) = 1$ needs to insert two nodes (4 and 6), and node $\psi^{-1}(4) = 4$ needs to insert one node (5). Since this is not possible in one step, the first step inserts only node 4 ($\nu_{1,1} = +1$) and the second step nodes 5 and 6 ($\nu_{2,1} = +1$, $\nu_{2,4} = +1$). The remaining edit scores ensure that the correct edges are inserted and deleted, yielding the final script at the bottom. This is exactly the script generated by $\psi$, except for four superfluous auxiliary edits, namely $\mathrm{eins}_{1,4}, \mathrm{eins}_{1,6}, \mathrm{edel}_{1,4}$, and $\mathrm{edel}_{1,6}$.

---

**Algorithm 2** An algorithm to turn a graph mapping $\psi$ between two graphs $G$ and $H$ into a teaching signal for a graph edit network, such that the teaching signal yields the same edit script as $\psi$, up to $2 \cdot (C-1)$ auxiliary edits, where $C$ is the number of connected components in $H$.

---

1: **function** MAP-TO-SIGNAL(Two non-empty graphs $G = (V = \{1, \ldots, M\}, E, \boldsymbol{X})$ and $H = (V' = \{1, \ldots, N\}, E', \boldsymbol{X'})$, a graph mapping $\psi$ between them)
2:      Compute the connected components of $H$ $\mathcal{C}_1, \ldots, \mathcal{C}_L$ where $\mathcal{C}_l \subseteq \text{Ins}_\psi$.
3:      Set $c_l \leftarrow \min_{j \in \mathcal{C}_l} j$ for all $l \in \{1, \ldots, L\}$.
4:      Set $r \leftarrow \min_{i \leq M : \psi(i) \leq N} i$.
5:      Construct $\tilde{H} = (V', E' \cup \{(\psi(r), c_1), \ldots, (\psi(r), c_L)\}), \boldsymbol{X'})$.
6:      **for** $j \in \text{Ins}_\psi$ **do**
7:          Compute a shortest path $\pi_1^j, \ldots, \pi_{R_j}^j$ from the closest node
8:          $\pi_1^j \in \{1, \ldots, N\} \setminus \text{Ins}_\psi$ to $\pi_{R_j}^j = j$ in $\tilde{H}$.
9:      **end for**
10:      **for** $k \in \{1, \ldots, |\text{Ins}_\psi|\}$ **do**
11:          Initialize $\vec{\nu}_k \leftarrow \vec{0}, \boldsymbol{Y}_k \leftarrow \boldsymbol{X}, \vec{e}_k^- \leftarrow -\vec{1}, \vec{e}_k^+ \leftarrow -\vec{1}, \boldsymbol{\mathcal{E}}_k \leftarrow \boldsymbol{0}$.
12:      **end for**
13:      Let $\text{ch}(j)$ be the children of node $j$ in the shortest path trees from roots $\pi_1^j$ to inserted nodes.
14:      Initialize a stack $\mathcal{S}$ with entries $(\pi_1^j, 0)$ for all $j \in \text{Ins}_\psi$. Set $K \leftarrow 0$.
15:      **while** $\mathcal{S}$ is not empty **do**
16:          Pop $(j, k)$ from $\mathcal{S}$. Set $c \leftarrow 1$. Set $i \leftarrow \psi^{-1}(j)$.
17:          **for** $j' \in \text{ch}(j)$ **do**
18:              Set $\nu_{k+c,i} \leftarrow +1$. $\vec{y}_{k+c,i} \leftarrow \vec{x}'_{j'}$.
19:              Add $(j', k+c)$ to $\mathcal{S}$.
20:              $c \leftarrow c + 1$. $K \leftarrow \max\{K, k+c\}$.
21:          **end for**
22:      **end while**
23:      Let $E''$ be the set of inserted edges $(i, j)$ due to line 18.
24:      Initialize $\vec{\nu}_{K+1} \leftarrow \vec{0}, \boldsymbol{Y}_{K+1} \leftarrow \boldsymbol{0}, \vec{e}_{K+1}^- \leftarrow -\vec{1}, \vec{e}_{K+1}^+ \leftarrow -\vec{1}, \boldsymbol{\mathcal{E}}_{K+1} \leftarrow \boldsymbol{0}$,
25:      Set $\vec{y}_{K+1,i} \leftarrow \vec{x}'_{\psi(i)}$ if $\psi(i) \leq N$.                  $\triangleright$ performs replacements
26:      Set $\epsilon_{K+1,i,j} \leftarrow +1$ if $(i, j) \notin E \cup E''$ and $(\psi(i), \psi(j)) \in E'$.    $\triangleright$ performs edge insertions
27:      Set $\epsilon_{K+1,i,j} \leftarrow -1$ if $(i, j) \in E \cup E''$ and $(\psi(i), \psi(j)) \notin E'$.    $\triangleright$ performs edge deletions
28:      Set $\nu_{K+1,i} \leftarrow -1$ if $\psi(i) > N$.                       $\triangleright$ performs node deletions
29:      Set $e_{K+1,i}^+ \leftarrow +1$ if $\epsilon_{K+1,i,j} \neq 0$ for any $j$.
30:      Set $e_{K+1,j}^- \leftarrow +1$ if $\epsilon_{K+1,i,j} \neq 0$ for any $i$.
31:      **return** $(\vec{\nu}_1, \boldsymbol{X}_1, \vec{e}_1^+, \vec{e}_1^-, \boldsymbol{\mathcal{E}}_1), \ldots, (\vec{\nu}_{K+1}, \boldsymbol{X}_{K+1}, \vec{e}_{K+1}^+, \vec{e}_{K+1}^-, \boldsymbol{\mathcal{E}}_{K+1})$.
32: **end function**

---

Lines 24-30 require only matrix and set lookups, making them overall $\mathcal{O}((M + N)^2)$. In summary, we obtain the claimed efficiency class of $\mathcal{O}(M^2 + N^2)$.

Now, let $G$ and $H$ be any two non-empty graphs, let $C$ be the number of connected components in $H$, and let $\psi$ be a graph mapping between them with $|\text{Ins}_\psi| < N$, *i.e.* at least one node is not inserted. Further, let $\bar{s} = (\hat{\nu}_1, \hat{\boldsymbol{X}}_1, \hat{e}_1^+, \hat{e}_1^-, \hat{\boldsymbol{\mathcal{E}}}_1), \ldots, (\hat{\nu}_{K+1}, \hat{\boldsymbol{X}}_{K+1}, \hat{e}_{K+1}^+, \hat{e}_{K+1}^-, \hat{\boldsymbol{\mathcal{E}}}_{K+1})$ be the teaching signal returned by Algorithm 2 for mapping $\psi$, and let $\bar{\delta}$ be the output of Algorithm 1 for input $\bar{s}$. Then, we wish to prove the following properties for $\bar{\delta}$.

1. $\bar{\delta}(G) \cong H$.
2. $|\bar{\delta}| \leq |\bar{\delta}_\psi| + 2 \cdot (C - 1)$.
3. The first $K$ steps of $\bar{s}$ contain only insertions with $\nu_{k,i} = 0 \Rightarrow \nu_{k+1,i} = \ldots \nu_{K+1,i} = 0$.
4. The $K + 1$th step contains no insertions.
5. $K \leq |\text{Ins}_\psi|$.

First, observe that lines 1-3 of Algorithm 2 are guaranteed to work because $H$ is non-empty and thus at least one connected component exists, and that each connected component is per definition non-empty. Further, line 4 works because we restricted $\psi$ such that at least one $i$ exists with $\psi(i) \leq N$, which means that the minimum is well-defined. Accordingly, the graph $\tilde{H}$ in line 5 is well-defined as well. This preparation was necessary to ensure that the shortest-path computations in lines 6-9 are valid. In particular, assume that there exists some $j \in \text{Ins}_\psi$ such that $j$ is disconnected in $\tilde{H}$ from any node $i \notin \text{Ins}_\psi$. Now, distinguish two cases. First, if $j$ lies in a connected component of $H$ which contains some node $i$ with $i \notin \text{Ins}_\psi$, then $j$ is connected to $i$, which directly yields a contradiction. Otherwise, $j$ lies in one of the connected components $\mathcal{C}_l$. Accordingly, $j$ is connected to $c_l$. However, then $j$ is also connected to $\psi(r)$ in $\tilde{H}$ by construction, where $r \notin \text{Ins}_\psi$, which is also a contradiction. Thus, the shortest path required in lines 6-9 is well-defined.

Next, observe that any shortest path computation gives rise to shortest path trees as required by line 13. If multiple shortest paths overlap, we obtain proper trees, otherwise trivial trees which are just copies of the shortest paths.

Lines 14 to 22 now insert all nodes in the shortest paths trees according to a particular ordering, namely that any parent is inserted at least one step before its child and each child is inserted one step before its right sibling. By using this ordering we ensure that no node makes more than one insertion per step, that whenever a node stops to do insertions, it never needs to start again, and that all nodes are reached (because the depth-first-search enumerates all nodes in the shortest path trees).

Accordingly, note that the $K$ first steps of the generates script, when plugged into Algorithm 1, yield exactly the insertions in $\bar{\delta}_\psi^{\text{ins}}$ as given in Definition 2 (up to reordering), in addition to edge insertions between an inserted node and its predecessor in the shortest paths tree. These inserted edges are stored in $E''$ in line 23.

The remainder of the algorithm takes care of all remaining edits. In particular, line 24 initializes a trivial teaching signal with zero node edit scores, zero attribute matrix, negative edge filter scores, and zero edit score matrix.

Then, line 25 sets all attributes to the target attributes, which yields all desired replacements of already existing nodes and leaves the attributes of inserted nodes as they were. In other words, line 25 yields, via Algorithm 1, exactly the script $\bar{\delta}_\psi^{\text{rep}}$ as given in Definition 2.

Line 26 sets edge insertion scores $\epsilon_{K+1,i,j} = 1$ for all edges $(i, j)$ such that $(\psi(i), \psi(j))$ is in $E'$ but not already in $E \cup E''$. In effect, after line 26 we have accumulated all edge insertions in $\bar{\delta}_\psi^{\text{eins}}$ as given in Definition 2, plus exactly $L$ edge insertions $\text{eins}_{r,\psi^{-1}(c_l)}$. This is the case because the shortest path to any node $c_l$ is necessarily $\pi_1^{c_l} = r$ and $\pi_2^{c_l} = c_l$. If any other, equally short path would exist, $c_l$ would be in a connected component with some node $i \notin \text{Ins}_\psi$, which is a contradiction. Accordingly, lines 14-22 have ensured that node $r$ inserts node $\psi^{-1}(c_l)$ at some point.

Next, line 27 sets edge deletion scores $\epsilon_{K+1,i,j} = -1$ for all edges $(i, j)$ that are in $E \cup E''$ but where $(\psi(i), \psi(j))$ is not in $E'$ anymore. Note that there is no overlap between these edges and the edges considered in line 26, such that we can handle both cases within the same step. Also note

that this step yields all edge deletions from from $\bar{\delta}_\psi^{\text{edel}}$ as given in Definition 2, plus exactly $L$ edge deletions $\text{edel}_{r,\psi^{-1}(c_l)}$ due to our previous argument.

Finally, line 28 sets node edit scores $\nu_{K+1,i} = -1$ for all nodes $i$ with $\psi(i) > N$. This yields exactly the node deletions from $\bar{\delta}_\psi^{\text{del}}$ as given in Definition 2.

Lines 29-30 merely ensure that the edge filter scores are consistent with $\mathcal{E}_k$.

In summary, when we plug the teaching signal returned by Algorithm 2 into Algorithm 1, the resulting script $\bar{\delta}$ contains exactly the same edits as $\bar{\delta}_\psi$ from Definition 2, up to reordering inserted nodes, plus $L$ edge insertions $\text{eins}_{r,\psi^{-1}(c_l)}$ and $L$ edge deletions $\text{edel}_{r,\psi^{-1}(c_l)}$. Because the edge deletions remove precisely the edges which have been additionally inserted before, we obtain $\bar{\delta}(G) \cong \bar{\delta}_\psi(G)$. Further, Lemma 1 shows that $\bar{\delta}_\psi(G) \cong H$, which implies property 1 above.

Next, observe that $L$ is exactly the number of connected components of $H$ which are subsets of $\text{Ins}_\psi$. Because we required that $\psi$ does not insert all nodes of $H$, there must exist at least one connected component of $H$ which is *not* a subset of $\text{Ins}_\psi$. Accordingly, $L$ is upper-bounded by $C - 1$ and we obtain the bound required in the second property.

For the third property, observe that the first $K$ steps do indeed only insert nodes. Further, a node $i$ only stops insertions when all its children in the shortest path tree are inserted. Once that is done, it does not start inserting anymore, implying the desired property.

The fourth property holds trivially because lines 24-30 never set a node score to a value $> \frac{1}{2}$.

Finally, the fifth property holds because each insertion step in lines 14-22 contains at least one insertion. This is because $K$ is set via line 20 exactly to a step in which the last insertion occurs and in insertion occuring in step $K$ implies that either a left sibling insertion or the parent insertion had to occur in step $K - 1$. However, if at least one insertion occurs in each step, there can be at most $K \leq |\text{Ins}_\psi|$ steps, as claimed.

It only remains to show that the bounds in properties 2 and 5 are sharp. To do so, consider the example graphs $G = (\{1\}, \emptyset, \mathbf{0})$ and $H = (\{1, \ldots, C\}, \emptyset, \mathbf{0})$. Note that all nodes in $H$ are isolated, such that they form trivial connected components. We note in passing that it is also possible to construct an example for non-trivial connected components, *e.g.* by constructing $H$ as $C$ copies of $G$, where $G$ can be arbitrarily shaped. Here, we consider the simplest case.

Now, let $\psi$ be any mapping between $G$ and $H$ such that $|\text{Ins}_\psi| < C$. In other words, $\psi(1) \leq C$. Then, line 2 of Algorithm 2 yields $\mathcal{C}_l = \{l\}$ for $l < \psi(1)$, and $\mathcal{C}_l = \{l + 1\}$ for $l \geq \psi(1)$ with $l \in \{1, \ldots, C - 1\}$ and $c_l$ becomes the only element of $\mathcal{C}_l$ in line 3. Line 4 yields $r = 1$. Accordingly, $\tilde{H} = (\{1, \ldots, C\}, \{(\psi(1), c_1), \ldots, (\psi(1), c_{C-1})\}, \mathbf{X}')$. Following the arguments above, the script resulting via Algorithm 1 from the output of Algorithm 2 then includes one edge insertion and one edge deletion for each edge in $\tilde{H}$, in addition to the $C - 1$ insertions that are contained in $\bar{\delta}_\psi$. Accordingly, we obtain $|\bar{\delta}| = |\bar{\delta}_\psi| + 2 \cdot (C - 1)$, which is precisely the bound. Further, because node 1 can only perform a single insertion per step we obtain $K = C - 1 = |\text{Ins}_\psi|$.

## B  AN EXAMPLE LOSS FUNCTION

For training, we propose a simple loss inspired by the margin hinge loss of the support vector machine (Suykens et al., 2002). Recall that the SVM hinge loss is given as

$$\ell^{\mathrm{SVM}}(\vec{w}, \{(\vec{x}, \hat{y})\}) = \sum_{(\vec{x}, \hat{y})} \left[ - \vec{w}^T \cdot \vec{x} \cdot \hat{y} + 1 \right]_+^2$$

where $\hat{y} \in \{-1, 1\}$ is the desired label and $y = \vec{w}^T \cdot \vec{x}$ is the predicted label.

In more detail, let $\boldsymbol{A}$ be the adjacency matrix of the input graph, let $\vec{\nu}, \boldsymbol{X}, \vec{e}^+, \vec{e}^-, \boldsymbol{\mathcal{E}}$ be the GEN layer output, and let $\hat{\nu}, \hat{\boldsymbol{X}}, \hat{e}^+, \hat{e}^-, \hat{\boldsymbol{\mathcal{E}}}$ be the scores of our teaching signal. Then, we define the loss $\ell_{\mathrm{GEN}}$ as:

$$\sum_{i=1}^{M} \ell^{\mathrm{node}}(\nu_i, \hat{\nu}_i) + \ell^{\mathrm{filter}}(e_i^+, \hat{e}_i^+) + \ell^{\mathrm{filter}}(e_i^-, \hat{e}_i^-) + \sum_{i:\hat{\nu}_i \geq -\frac{1}{2}} \ell^{\mathrm{attr}}(\vec{x}_i, \hat{x}_i) + \sum_{(i,j):\hat{e}_i^+>0, \hat{e}_j^->0} \ell^{\mathrm{edge}}(\epsilon_{ij}, \hat{\epsilon}_{ij}, a_{ij}),$$

(2)

where $\ell^{\mathrm{node}}(\nu, \hat{\nu})$ is defined as $[\nu + 1]_+^2$ for $\hat{\nu} < -\frac{1}{2}$, as $[-\nu + 1]_+^2$ for $\hat{\nu} > +\frac{1}{2}$, and as $[|\nu| - \frac{1}{4}]_+^2$ otherwise, with $[x]_+ = \max\{0, x\}$ being the rectified linear unit; where $\ell^{\mathrm{filter}}(e, \hat{e})$ is defined as $M \cdot [e + \frac{1}{2}]_+^2$ if $\hat{e} \leq 0$ and as $M \cdot [-e + \frac{1}{2}]_+^2$ otherwise; where $\ell^{\mathrm{attr}}$ is the squared Euclidean distance for continuous attributes and crossentropy for discrete attributes; and where $\ell^{\mathrm{edge}}(\epsilon, \hat{e}, a)$ is defined as $[\epsilon + 1]_+^2$ if $a = 1$ and $\hat{e} < -\frac{1}{2}$, as $[-\epsilon]_+^2$ if $a = 1$ and $\hat{e} \geq -\frac{1}{2}$, as $[\epsilon]_+^2$ if $a = 0$ and $\hat{e} \leq \frac{1}{2}$, and as $[-\epsilon + 1]^2$ if $a = 0$ and $\hat{e} > \frac{1}{2}$. We can show that this loss is zero if and only if the scripts of the teaching signal and the GEN output are equal (plus a loss for margin violations).

**Theorem 3.** *Let* $\boldsymbol{A} \in \{0,1\}^{N \times N}$ *be an adjacency matrix and* $\vec{\nu}, \boldsymbol{X}, \vec{e}^+, \vec{e}^-, \boldsymbol{\mathcal{E}}$ *as well as* $\hat{\nu}, \hat{\boldsymbol{X}}, \hat{e}^+, \hat{e}^-, \hat{\boldsymbol{\mathcal{E}}}$ *be two sets of edit scores.*

*Then, loss 2 is zero if and only if the following conditions hold.*

1. *For all* $i$ *: If* $\hat{\nu}_i > +\frac{1}{2}$, $\nu_i$ *is at least 1,*

2. *for all* $i$ *: if* $|\hat{\nu}_i| \leq \frac{1}{2}$, $|\nu_i|$ *is at most* $\frac{1}{4}$,

3. *for all* $i$ *: if* $\hat{\nu}_i < -\frac{1}{2}$, $\nu_i$ *is at most* $-1$,

4. *for all* $i$ *: If* $\hat{\nu}_i \geq -\frac{1}{2}$, $\ell^{\mathrm{attr}}(\vec{x}_i, \hat{x}_i) = 0$,

5. *for all* $i$ *: If* $\hat{e}_i^+ > 0$, $e_i^+$ *is at least* $\frac{1}{2}$,

6. *for all* $i$ *: If* $\hat{e}_i^+ \leq 0$, $e_i^+$ *is at most* $-\frac{1}{2}$,

7. *for all* $i$ *: If* $\hat{e}_i^- > 0$, $e_i^-$ *is at least* $\frac{1}{2}$,

8. *for all* $i$ *: If* $\hat{e}_i^- \leq 0$, $e_i^-$ *is at most* $-\frac{1}{2}$,

9. *for all* $i, j$ *: If* $\hat{e}_i^+ > 0$, $\hat{e}_j^- > 0$, $\hat{\epsilon}_{i,j} > +\frac{1}{2}$, *and* $a_{i,j} = 0$, $\epsilon_{i,j}$ *is at least 1,*

10. *for all* $i, j$ *: If* $\hat{e}_i^+ > 0$, $\hat{e}_j^- > 0$, $\hat{\epsilon}_{i,j} \leq \frac{1}{2}$, *and* $a_{i,j} = 0$, $\epsilon_{i,j}$ *is at most 0,*

11. *for all* $i, j$ *: If* $\hat{e}_i^+ > 0$, $\hat{e}_j^- > 0$, $\hat{\epsilon}_{i,j} \geq -\frac{1}{2}$, *and* $a_{i,j} = 1$, $\epsilon_{i,j}$ *is at least 0, and*

12. *for all* $i, j$ *: If* $\hat{e}_i^+ > 0$, $\hat{e}_j^- > 0$, $\hat{\epsilon}_{i,j} < -\frac{1}{2}$, *and* $a_{i,j} = 1$, $\epsilon_{i,j}$ *is at most* $-1$.

*Further, if the loss is zero, the edit scripts resulting from Algorithm 1 for both sets of edits are equal.*

*Proof.* We first re-frame all conditions in terms of the component losses $\ell^{\mathrm{node}}$, $\ell^{\mathrm{attr}}$, $\ell^{\mathrm{filter}}$, and $\ell^{\mathrm{edge}}$.

1. In this case, the node loss is $\ell^{\mathrm{node}}(\nu_i, \hat{\nu}_i) = \left[ -\nu_i + 1 \right]_+^2$. Accordingly, the loss is zero iff $-\nu_i + 1 \leq 0 \iff \nu_i \geq 1$, as claimed.

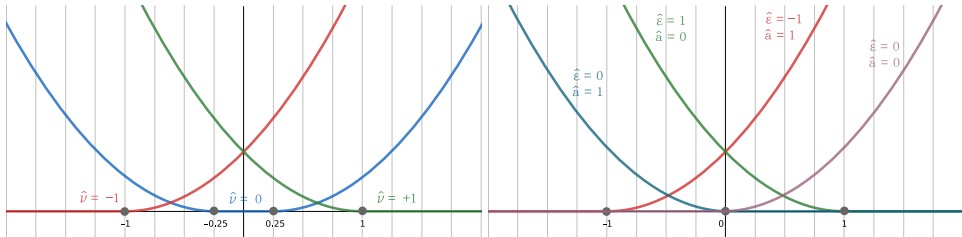

Figure 5: An illustration of the loss $\ell^{\mathrm{node}}$ (left) and the loss $\ell^{\mathrm{edge}}$ (right) for different teacher and adjacency matrix inputs.

2. In this case, the node loss is $\ell^{\mathrm{node}}(\nu_i, \hat{\nu}_i) = \left[|\nu_i| - \frac{1}{4}\right]_+^2$. Accordingly, the loss is zero iff $|\nu_i| - \frac{1}{4} \leq 0 \iff |\nu_i| \leq \frac{1}{4}$, as claimed.

3. In this case, the node loss is $\ell^{\mathrm{node}}(\nu_i, \hat{\nu}_i) = \left[\nu_i + 1\right]_+^2$. Accordingly, the loss is zero iff $\nu_i + 1 \leq 0 \iff \nu_i \leq -1$, as claimed.

4. This condition directly refers to the attribute loss and is obvious.

5. In this case, the filter loss is $\ell^{\mathrm{filter}}(e_i^+, \hat{e}_i^+) = \left[-e_i^+ + \frac{1}{2}\right]_+^2$. Accordingly, the loss is zero iff $-e_i^+ + \frac{1}{2} \leq 0 \iff e_i^+ \geq \frac{1}{2}$, as claimed.

6. In this case, the filter loss is $\ell^{\mathrm{filter}}(e_i^+, \hat{e}_i^+) = \left[e_i^+ + \frac{1}{2}\right]_+^2$. Accordingly, the loss is zero iff $e_i^+ + \frac{1}{2} \leq 0 \iff e_i^+ \leq -\frac{1}{2}$, as claimed.

7. In this case, the filter loss is $\ell^{\mathrm{filter}}(e_i^-, \hat{e}_i^-) = \left[-e_i^- + \frac{1}{2}\right]_+^2$. Accordingly, the loss is zero iff $-e_i^- + \frac{1}{2} \leq 0 \iff e_i^- \geq \frac{1}{2}$, as claimed.

8. In this case, the filter loss is $\ell^{\mathrm{filter}}(e_i^-, \hat{e}_i^-) = \left[e_i^- + \frac{1}{2}\right]_+^2$. Accordingly, the loss is zero iff $e_i^- + \frac{1}{2} \leq 0 \iff e_i^- \leq -\frac{1}{2}$, as claimed.

9. In this case, the edge loss is $\ell^{\mathrm{edge}}(\epsilon_{i,j}, \hat{\epsilon}_{i,j}, a_{i,j}) = \left[-\epsilon_{i,j} + 1\right]_+^2$. Accordingly, the loss is zero iff $-\epsilon_{i,j} + 1 \leq 0 \iff \epsilon_{i,j} \geq 1$, as claimed.

10. In this case, the edge loss is $\ell^{\mathrm{edge}}(\epsilon_{i,j}, \hat{\epsilon}_{i,j}, a_{i,j}) = \left[\epsilon_{i,j}\right]_+^2$. Accordingly, the loss is zero iff $\epsilon_{i,j} \leq 0$, as claimed.

11. In this case, the edge loss is $\ell^{\mathrm{edge}}(\epsilon_{i,j}, \hat{\epsilon}_{i,j}, a_{i,j}) = \left[-\epsilon_{i,j}\right]_+^2$. Accordingly, the loss is zero iff $-\epsilon_{i,j} \leq 0 \iff \epsilon_{i,j} \geq 0$, as claimed.

12. In this case, the edge loss is $\ell^{\mathrm{edge}}(\epsilon_{i,j}, \hat{\epsilon}_{i,j}, a_{i,j}) = \left[\epsilon_{i,j} + 1\right]_+^2$. Accordingly, the loss is zero iff $\epsilon_{i,j} + 1 \leq 0 \iff \epsilon_{i,j} \leq -1$, as claimed.

Accordingly, we found that all component losses are zero if and only if the aforementioned margin conditions holds. Since the entire loss 2 is a sum of these component losses and no contribution can be negative in this case, the entire sum is zero if and only if all contributions are zero, which proves the claim. Also refer to Figure 5 for a graphical illustration of the component losses $\ell^{\mathrm{node}}$ and $\ell^{\mathrm{edge}}$.

This directly yields the second part of Theorem 3. In particular, if condition 3 holds, the same edge deletions are appended in line 2 of Algorithm 1. Due to conditions 5-8, the loops in lines 3-8 iterate over the same elements $(i, j)$. Further, due to conditions 11-12, Algorithm 1 appends the same edge deletions in line 5, and due to conditions 9-10, Algorithm 1 appends the same edge insertions in line 6. The replacements in line 9 are the same due to conditions 2 and 4, as well as the fact that $\ell^{\mathrm{attr}}$ being zero implies equal inputs by definition. The loop in lines 11-14 now iterates over the same elements due to condition 1, and the same node/edge insertions are appended in line 12 due to

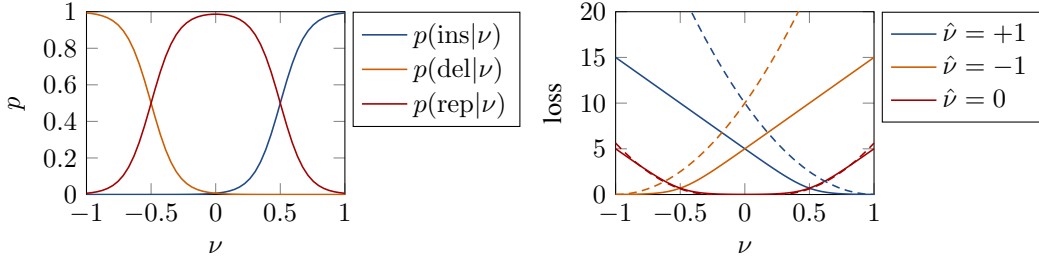

Figure 6: Left: The node edit probabilities $p(\text{ins}|\nu)$ (blue), $p(\text{del}|\nu)$ (orange), and $p(\text{rep}|\nu)$ (red) for $\beta = 10$ for varying $\nu$ (left). Right: The crossentropy loss (solid) and the GEN loss times $\beta$ (dashed) for varying $\nu$ and for $\hat{\nu} = +1$ (blue), $\hat{\nu} = -1$ (orange), and $\hat{\nu} = 0$ (red).

condition 4 and the fact that $\ell^{\text{attr}}$ being zero implies equal inputs by definition. Finally, the same node deletions are generated in line 15 due to condition 3. Thus, the output of Algorithm 1 is the same for both inputs. $\qquad\square$

## C    PROBABILISTIC INTERPRETATION OF THE GEN

Given the node edit scores $\nu_i$, the edge filter scores $e_i^+$ and $e_i^-$ as well as the edge edit scores $\epsilon_{i,j}$ as returned by a GEN layer, we can define the following probabilities over edits. The probability of a node insertion at node $i$ is the logistic distribution $p(\text{ins}|\nu_i) = Z_\beta/(1 + \exp[-\beta \cdot (\nu_i - \frac{1}{2})])$; the probability of a deletion of node $i$ is the logistic distribution $p(\text{del}|\nu_i) = Z_\beta/(1 + \exp[-\beta \cdot (-\nu_i - \frac{1}{2})])$; and the probability of a replacement is $p(\text{rep}|\nu_i) = 1 - p(\text{ins}|\nu_i) - p(\text{del}|\nu_i)$. In all cases, $\beta > 0$ is a hyper-parameter regulating the slope of the distribution with respect to $\nu_i$ and $Z_\beta = (1 + e^\beta)/(2 + e^\beta)$ is chosen to ensure that $p(\text{ins}|\frac{1}{2}) = p(\text{rep}|\frac{1}{2}) = p(\text{del}|-\frac{1}{2}) = p(\text{rep}|-\frac{1}{2})$, *i.e.* $\frac{1}{2}$ is the decision boundary between insertion and replacement and $-\frac{1}{2}$ is the decision boundary between deletion and replacement, just as in Algorithm 1. For sufficiently high $\beta$, *e.g.* $\beta \geq 10$, we obtain $p(\text{ins}|\nu_i) \approx 0$ for $\nu_i < 0$ and $p(\text{del}|\nu_i) \approx 0$ for $\nu_i > 0$, such that the distribution becomes equivalent to two binary logistic distributions 'glued together'. Also refer to Figure 6 (left).

A straightforward loss based on this probability distribution is the crossentropy between $p$ and the strict teaching distribution $q(\text{ins}|\hat{\nu}_i) = 1$ if $\hat{\nu}_i > \frac{1}{2}$, $q(\text{del}|\hat{\nu}_i)$ if $\hat{\nu}_i < -\frac{1}{2}$, and $q(\text{rep}|\hat{\nu}_i) = 1$ otherwise. The loss for varying $\nu$ and $\hat{\nu}$ and $\beta = 10$ is shown in Figure 6 (right). As we can see, the behavior is qualitatively very similar to the GEN loss from Equation 2, shown in dashed lines. The most notable differences are that the crossentropy behaves linearly for sufficiently extreme $\nu$, whereas the GEN loss behaves quadratic, and that the GEN loss is strictly zero when the margin is not violated, whereas the crossentropy still remains strictly larger than zero. This can lead to empiric problems during learning, as we see in the experiments. The quadratic behavior emphasizes large margin violations whereas small margin violations are less emphasized compared to the crossentropy loss, which yields slightly favorable properties in our experiments.

To sample edge edits we perform a three-step process. First, for each node we sample whether outgoing edges or incoming edges are edited with the logistic distribution $p(\top|e_i^+) = 1/(1 + \exp[-\beta \cdot e_i^+])$ and $p(\top|e_i^-) = 1/(1 + \exp[-\beta \cdot e_i^-])$. Then, we only consider edges $(i, j)$ where $\top$ has been sampled for both $i$ and $j$, and we sample an edge insertion with probability $p(\text{eins}|\epsilon_{i,j}, a_{i,j}) = 1/(1 + \exp[-\beta \cdot (\epsilon_{i,j} - \frac{1}{2})])$ if $a_{i,j} = 0$ and $p(\text{eins}|\epsilon_{i,j}, a_{i,j}) = 0$ if $a_{i,j} = 1$, and we sample an egde deletion with probability $p(\text{edel}|\epsilon_{i,j}, a_{i,j}) = 1/(1 + \exp[-\beta \cdot (-\epsilon_{i,j} - \frac{1}{2})])$ if $a_{i,j} = 1$ and $p(\text{eins}|\epsilon_{i,j}, a_{i,j}) = 0$ if $a_{i,j} = 0$. Also refer to Figure 7 (left).

The crossentropy loss compared to the strict distribution $q(\text{eins}|\hat{\epsilon}_{i,j}) = 1$ if $\hat{\epsilon}_{i,j} > \frac{1}{2}$, $q(\text{edel}|\hat{\epsilon}_{i,j}) = 1$ if $\hat{\epsilon}_{i,j} < -\frac{1}{2}$, and $q(\text{eins}|\hat{\epsilon}_{i,j}) = q(\text{edel}|\hat{\epsilon}_{i,j}) = 0$ is shown in Figure 7 (right). Again, we observe that the loss behaves qualitatively very similar to the GEN edge loss, with the striking difference being the square.

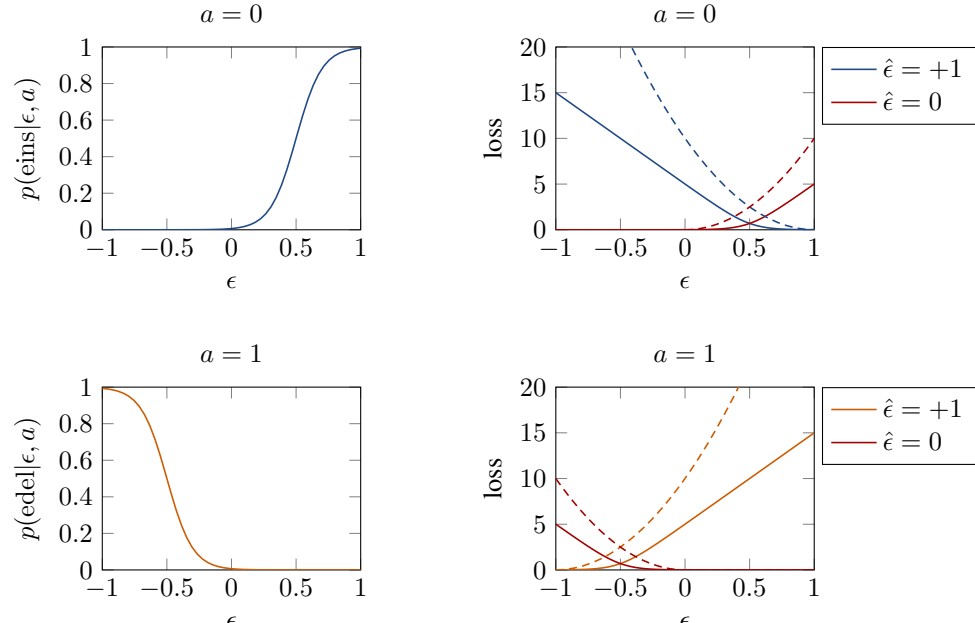

Figure 7: Left top: The edge insertion probability $p(\text{eins}|\epsilon, a)$ if the edge is not present. Right top: The crossentropy loss (solid) and the GEN edge loss times $\beta$ (dashed) for $\hat{\nu} = +1$ (blue) and $\hat{\nu} = 0$ (red) if the edge is not present. Left bottom: The edge deletion probability $p(\text{edel}|\epsilon, a)$ if the edge is present. Right bottom: The crossentropy loss (solid) and the GEN edge loss times $\beta$ (dashed) for $\hat{\nu} = -1$ (orange) and $\hat{\nu} = 0$ (red) if the edge is present.

Table 2: Summary statistics for all synthetic datasets, in particular the number of unique graphs, the average time series length $\bar{T}$ ($\pm$ standard deviation) and the average graph size $\bar{N}$ ($\pm$ standard deviation). The number of unique undirected graphs with 8 nodes for degree rules was computed using the `nauty-geng` function, all averages were obtained via 1000 samples from the dataset.

|  | edit cycles | degree rules | game of life | Boolean | Peano |
|---|---|---|---|---|---|
| unique graphs | 9 | 12346 | $2^{100}$ | 10788 | 34353 |
| $\bar{T}$ | $3 \pm 0.82$ | $10.46 \pm 1.91$ | $10 \pm 0.00$ | $1.51 \pm 0.99$ | $11.08 \pm 11.63$ |
| $\bar{N}$ | $3.22 \pm 1.03$ | $7.33 \pm 1.98$ | $100 \pm 0.00$ | $4.84 \pm 2.52$ | $7.82 \pm 3.89$ |

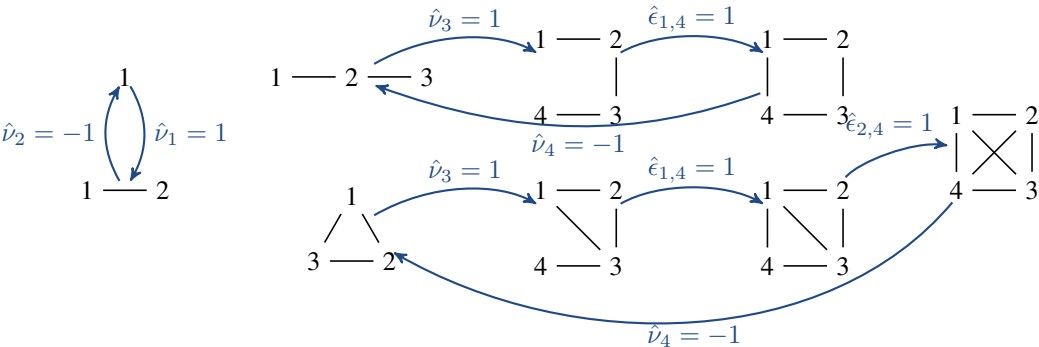

Figure 8: An illustration of the edit cycles dataset, where each graph is shown in black, whereas colored arrows illustrate the system dynamics. Each colored arrow is labelled with the edit scores necessary to move from one graph to the next.

## D    DATA IN DETAIL

In this section, we discuss the graph dynamical systems of Section 4 in more detail and also discuss how graph edit networks can realize a solution. The summary statistics are shown in Table 2.

**Edit Cycles:**  The edit cycles dataset is illustrated in Figure 8. Colored, thick arrows indicate the simulated system dynamics. Each arrow is labelled with the corresponding teaching protocol.

The reference solution to this task is to uniquely identify at each node to which graph it belongs and then decide on the edit. This requires a number of layers equal to the graph diameter, in this case two graph neural network layers.

**Degree Rules:**  The degree rules dataset is illustrated for two example graphs in Figure 2. The teaching protocol implements the rules of the dataset as follows. First, for any node $i$ with degree larger than 3 we expect $\hat{\nu}_i < -\frac{1}{2}$, *i.e.* the node should get deleted. Second, for any pair of nodes $i$ and $j$ which share at least one neighbor, we expect $\hat{\epsilon}_{i,j} = \hat{\epsilon}_{j,i} = 1$, *i.e.* an edge should be inserted between the nodes. Third, for any node $i$ with degree smaller than 3 we expect $\hat{\nu}_i > \frac{1}{2}$, *i.e.* a node should get inserted.

To generate the actual dynamics, we limit the process to apply only one rule per connected component. This is because simultaneous application of all rules leads to degeneracies. For example, a 5-clique would be deleted entirely. Instead, we use the following ordering: We first apply rule 1 to the node with highest degree. If rule 1 applies to no node, we apply rule 2 to the node pair with lowest sum of degrees. If rule 2 does not apply, we apply rule 3 to the node with lowest degree. If multiple nodes have the same degree, nodes with lower index take precedence. If no rule applies, we perform no edit and the process has converged.

Next, we show that each connected component converges to a 4-clique in this scheme. Note that rule 2 ensures that every connected component becomes fully connected, provided that rule 1 does not interfere. Rule 1 interferes if a node degree rises above 3, which will be the case whenever a connected component contains more than 4 nodes. Accordingly, every connected component with

more than 4 nodes must converge to a 4-clique. For smaller connected components, rule 2 first ensures that the component becomes a $k$-clique for $k < 4$. Then, rule 2 does not apply anymore but rule 3 applies because all nodes in the component have degree less than 3. Once a new node has been added, rule 2 applies again, and so on, until the component is a 4-clique. Finally, note that a 4-clique is stable because no rule applies anymore.

The reference solution for this task is a one-layer net with $n + 2$ neurons. Further, we assign a one-hot coding of the node index as attribute. Note that the ordering does not matter, merely that every node has a unique one-hot code. Now, notice that we can compute the degree of node $i$ via the expression $\sum_{j \in \mathcal{N}(i)} \vec{1}^T \cdot \vec{x}_j$. This is the case because $\vec{x}_j$ is a one-hot coding of the index $j$ and, hence, the expression $\vec{1}^T \cdot \vec{x}_j$ is always 1, such that the sum just counts the size of the neighborhood of $i$, which is exactly the degree. Accordingly, we need to apply rule 1 if this expression is higher than 3, and we need to apply rule 3 if this expression is less than 3. Rule 2 is slightly more complicated. We wish to apply rule 2 whenever nodes $i$ and $j$ share at least one neighbor, i.e. if $\mathcal{N}(i) \cap \mathcal{N}(j) \neq \emptyset$. Because the node attributes are one-hot codings of the node index, we can represent the neighborhoods by the sum of the one-hot codings in the neighborhood and the set intersection by their inner product. More precisely, we obtain for any two nodes $i$ and $j$:

$$\Big( \sum_{k \in \mathcal{N}(i)} \vec{x}_k \Big)^T \cdot \Big( \sum_{l \in \mathcal{N}(j)} \vec{x}_l \Big) = |\mathcal{N}(i) \cap \mathcal{N}(j)|.$$

Based on this result, we can identify the need to apply any of the three rules by setting $f^1_{\text{aggr}}$ and $f^1_{\text{merge}}$ from Equation 1 as follows.

$$f^1_{\text{aggr}}(\mathcal{N}) = \sum_{j \in \mathcal{N}} \vec{x}_j, \quad \text{and} \quad f^1_{\text{merge}}(\vec{x}, \vec{y}) = \text{ReLU} \begin{pmatrix} \vec{y} \\ \frac{1}{\beta} \cdot (\vec{1}^T \cdot \vec{y} - 3) \\ -\frac{1}{\beta} \cdot (\vec{1}^T \cdot \vec{y} - 2) \end{pmatrix},$$

which yields the following representation for all nodes $i$:

$$\phi^1(i) = \text{ReLU} \begin{pmatrix} \sum_{j \in \mathcal{N}(i)} \vec{x}_j \\ \frac{1}{\beta} \cdot (|\mathcal{N}(i)| - 3) \\ -\frac{1}{\beta} \cdot (|\mathcal{N}(i)| - 3) \end{pmatrix}$$

Here, $\beta$ is defined as $\beta := 2 \cdot \max\{3, \hat{m}\}$, where $\hat{m}$ is the maximum degree in the dataset. Then, we can compute the node edit score as $\nu_i = \beta \cdot (\phi^1(i)_{n+2} - \phi^1(i)_{n+1})$, which is $\geq 1$ if and only if the degree of $i$ is smaller than 3 (rule 3), which is $\leq -1$ if and only if the degree of $i$ is larger than 3 (rule 1), and which is zero otherwise. Further, we can set the edge edit score as $\epsilon_{i,j} = \phi^1(i)^T \cdot \phi^1(j)$, which is $\geq 1$ if $|\mathcal{N}(i) \cap \mathcal{N}(j)| \geq 1$ and which is between 0 and $\frac{1}{2}$ otherwise because in that case

$$\begin{aligned} \phi^1(i)^T \cdot \phi^1(j) = &\frac{1}{\beta^2} \cdot \big[ \text{ReLU}(|\mathcal{N}(i)| - 3) \cdot \text{ReLU}(|\mathcal{N}(j)| - 3) \\ &+ \text{ReLU}(3 - |\mathcal{N}(i)|) \cdot \text{ReLU}(3 - |\mathcal{N}(j)|) \big] \\ \leq &\frac{1}{\beta^2} \cdot (2 \cdot \max\{3, \hat{m}\}^2) \leq \frac{1}{2} \end{aligned}$$

Still, note that this is relatively difficult to learn for a neural net because parameters need to be set to relatively large values (exceeding the maximum degree). Hence the long experimental learning times.

**Game of life:** Conway's game of life is a 2D cellular automaton with the following rules. Cells can be either alive ($x^t_i = 1$) or dead ($x^t_i = 0$) at time $t$. The state in time $t + 1$ is given by the equation

$$x^{t+1}_i = \begin{cases} 1 & \text{if } 5 \leq x^t_i + 2 \cdot \sum_{j \in \mathcal{N}(i)} x^t_j \leq 7 \\ 0 & \text{otherwise} \end{cases}$$

where $\mathcal{N}(i)$ is the 8-neighborhood of $i$ in the grid. Figure 9 shows the development of an example grid.

Our teaching protocol imposes $\hat{\nu}^t_i = x^{t+1}_i - x^t_i$ for all nodes $i$ and all times $t \geq 1$, *i.e.* the node edit score $\nu^t_i$ should correspond to the change in 'aliveness'. Note that our graph connects all neighboring nodes in the grid and that the aliveness $x^t_i$ is the only node attribute.

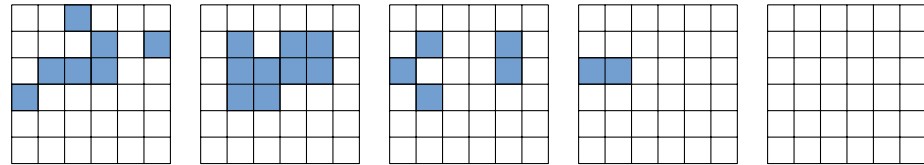

Figure 9: An example $6 \times 6$ grid evolving over time according to the rules of Conway's game of life with alive cells marked blue and dead cells white. Note that even a single alive cell close to a 'glider' shape breaks the shape and leads to the entire board becoming empty over time.

Our reference solution is a one-layer graph neural network with the following setting of $f^1_{\text{aggr}}$ and $f^1_{\text{merge}}$ from Equation 1.

$$f^1_{\text{aggr}}(\mathcal{N}) = \sum_{j \in \mathcal{N}} \vec{x}_j \quad \text{and} \quad f^1_{\text{merge}}(x, y) = \text{ReLU} \begin{pmatrix} x + 2y - 4 \\ x + 2y - 5 \\ x + 2y - 7 \\ x + 2y - 8 \\ x \end{pmatrix}.$$

Now, let $y_i := x_i + 2 \sum_{j \in \mathcal{N}(i)} x_j$. Then, we obtain the node representation $\phi^1(i) = \text{ReLU}(y_i - 4, y_i - 5, y_i - 7, y_i - 8, x_i)^T$.

With this representation, we obtain the following result for $\phi^1_1(i) - \phi^1_2(i) - \phi^1_3(i) + \phi^1_4(i)$.

$$\phi^1_1(i) - \phi^1_2(i) - \phi^1_3(i) + \phi^1_4(i) = \begin{cases} 0 - 0 - 0 + 0 = 0 & \text{if } y_i \leq 4 \\ y_i - 4 - (y_i - 5) - 0 + 0 = 1 & \text{if } 5 \leq y_i \leq 7 \\ y_i - 4 - (y_i - 5) - (y_i - 7) + y_i - 8 = 0 & \text{if } y_i \geq 8 \end{cases}$$

In other words, we obtain $\phi^1_1(i) - \phi^1_2(i) - \phi^1_3(i) + \phi^1_4(i) = x^{t+1}_i$. Accordingly, we can simply set $\nu_i = \phi^1_1(i) - \phi^1_2(i) - \phi^1_3(i) + \phi^1_4(i) - \phi^1_5(i)$ and obtain $\nu_i = x^{t+1}_i - x^t_i$ as desired.

**Boolean Formulae:** We first generate a random Boolean formula using the following stochastic process: Let $\Sigma = \{\wedge, \vee, x, y, \neg x, \neg y, \top, \bot, \text{root}\}$ be our alphabet of symbols. Then, we initialize a counter $\text{ops} \leftarrow 0$, a graph $G = (\{1\}, \emptyset, \boldsymbol{X})$ with $\vec{x}_1 = \text{root}$, and a stack with 1 as single element. Next, as long as the stack is not empty, we pop the top node $i$ from the stack and sample a new symbol from $\Sigma$ with probabilities $P(\wedge) = P(\vee) = 0.3$, $P(x) = P(y) = P(\neg x) = P(\neg y) = 0.1$, and $P(\text{root}) = P(\top) = P(\bot) = 0$. Then, we add a new node $M + 1$ to the graph with $\vec{x}_{M+1}$ being the one-hot-code of the samples symbol, and add the edge $(i, M + 1)$. Further, if we have sampled $\wedge$ or $\vee$, we push $i$ onto the stack two times and increment the $\text{ops}$ counter. If it reaches 3, we adjust the sampling probabilities to $P(x) = P(y) = P(\neg x) = P(\neg y) = \frac{1}{4}$ and $P(\wedge) = P(\vee) = P(\text{root}) = P(\top) = P(\bot) = 0$.

Once the initial graph is sampled, the teaching protocol implements eight simplification rules for Boolean formulae. In particular, we implement the following rules.

1. $F \wedge \bot \iff \bot$ for any formula $F$. Accordingly, if for two nodes $i, j$ we have $\vec{x}_i = \wedge$, $\vec{x}_j = \bot$ and $(i, j) \in E$, we impose $\hat{\nu}_k = -1$ for any node $k \neq j$ with $(i, k) \in E$. As a minor side note, we mention that we consider tree edits according to Zhang & Shasha (1989) for this dataset instead of graph edits, i.e. when we delete node $i$, the children of $i$ automatically get connected to the parent of $i$. If $F$ is a leaf, we also impose $\hat{\nu}_i = -1$.

2. $F \wedge \top \iff F$ for any $F$. Accordingly, if for two nodes $i, j$ we have $\vec{x}_i = \wedge$, $\vec{x}_j = \top$ and $(i, j) \in E$, we impose $\hat{\nu}_i = \hat{\nu}_j = -1$.

3. $F \vee \top \iff \top$ for any $F$. Accordingly, we apply the same scheme as in rule 1, just with the condition $\vec{x}_i = \vee$ and $\vec{x}_j = \top$.

4. $F \vee \bot \iff F$ for any $F$. Accordingly, if for two nodes $i, j$ we have $\vec{x}_i = \vee$, $\vec{x}_j = \bot$ and $(i, j) \in E$, we impose $\hat{\nu}_i = \hat{\nu}_j = -1$.

5. $x \wedge x \iff x$ and $y \wedge y \iff y$. Accordingly, if we find three nodes $i, j, k$, such that $\vec{x}_i = \wedge$, $\vec{x}_j = \vec{x}_k \in \{x, y\}$, and $(i, j), (i, k) \in E$, we impose $\hat{\nu}_i = \hat{\nu}_j = -1$.

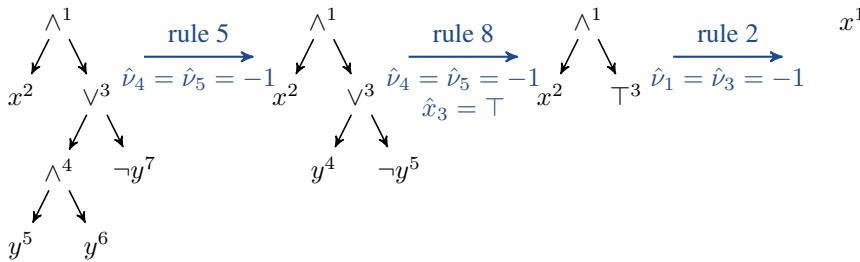

Figure 10: An example time series from the Boolean dataset. The initial formula on the left is simplified until no simplification rule applies anymore. Each arrow indicates one simplification step. Superscripts indicate the index of each node in the graph and blue arrow labels indicate the teaching protocol.

6. $x \vee x \iff x$ and $y \vee y \iff y$. Accordingly, if we find three nodes $i, j, k$, such that $\vec{x}_i = \vee, \vec{x}_j = \vec{x}_k \in \{x, y\}$, and $(i, j), (i, k) \in E$, we impose $\hat{\nu}_i = \hat{\nu}_j = -1$.

7. $x \wedge \neg x \iff \bot$ and $y \wedge \neg y \iff \bot$. Accordingly, if we find three nodes $i, j, k$, such that $\vec{x}_i = \wedge, (i, j), (i, k) \in E$, and $\vec{x}_j = x$ as well as $\vec{x}_k = \neg x$, or $\vec{x}_j = y$ as well as $\vec{x}_k = \neg y$, we impose $\hat{\nu}_j = \hat{\nu}_k = -1$ and $\hat{y}_i = \bot$.

8. $x \vee \neg x \iff \top$ and $y \vee \neg y \iff \top$. Accordingly, if we find three nodes $i, j, k$, such that $\vec{x}_i = \vee, (i, j), (i, k) \in E$, and $\vec{x}_j = x$ as well as $\vec{x}_k = \neg x$, or $\vec{x}_j = y$ as well as $\vec{x}_k = \neg y$, we impose $\hat{\nu}_j = \hat{\nu}_k = -1$ and $\hat{y}_i = \top$.

The next step in the dynamic is the graph that results from applying all edits imposed by the teaching protocol. The dynamic process ends once no rules apply anymore. An example simplification process is shown in Figure 10.

The reference solution is a graph neural network with two layers, where the first layer represents whether certain conditions for the eight rules apply and the second layer implements conjunctions between these conditions. For simplicity, we only consider rule 2 here but all other rules can be implemented with a similar scheme. In particular, let $\phi(x)$ be the one-hot coding of the symbol $x \in \Sigma$. Then, we use the following setting of $f^1_{\text{aggr}}, f^1_{\text{merge}}, f^2_{\text{aggr}}$, and $f^2_{\text{merge}}$ from Equation 1.

$$f^1_{\text{aggr}}(\mathcal{N}^+, \mathcal{N}^-) = \begin{pmatrix} \sum_{j \in \mathcal{N}^+} \vec{x}_j \\ \sum_{j \in \mathcal{N}^-} \vec{x}_j \end{pmatrix} \quad \text{and} \quad f^1_{\text{merge}}\left(\vec{x}, \begin{pmatrix} \vec{y} \\ \vec{z} \end{pmatrix}\right) = \text{ReLU} \begin{pmatrix} \phi(\wedge)^T \cdot \vec{x} \\ \phi(\top)^T \cdot \vec{y} \\ \phi(\top)^T \cdot \vec{x} \\ \phi(\wedge)^T \cdot \vec{z} \end{pmatrix}$$

$$f^2_{\text{aggr}}(\mathcal{N}^+, \mathcal{N}^-) = 0 \quad \text{and} \quad f^2_{\text{merge}}(\vec{\phi}^1, 0) = \text{ReLU} \begin{pmatrix} \phi^1_1 + \phi^1_2 - 1 \\ \phi^1_3 + \phi^1_4 - 1 \end{pmatrix}$$

Note that we distinguish the neighborhood $\mathcal{N}^+$ of children and $\mathcal{N}^-$ of parents. The resulting node representation in layer 2 is as follows. $\phi^2_1(i) = 1$ if $\vec{x}_i = \wedge$ and $i$ has a child $j$ with $\vec{x}_j = \top$. Otherwise, $\phi^2_1(i) = 0$. Further, $\phi^2_2(i) = 1$ if $\vec{x}_i = \top$ and $i$ has a parent $j$ with $\vec{x}_j = \top$. Otherwise, $\phi^2_2(i) = 0$. Accordingly, we can implement rule 2 by setting $\nu_i = -\phi^2_1(i) - \phi^2_2(i)$, which deletes both the parent and the child if rule 2 applies.

**Peano addition:** With the same scheme as for the Boolean dataset we first generate a random addition with at most 3 plus operators. However, for the peano dataset we have the alphabet $\Sigma = \{+, 0, 1, \dots, 9, \text{succ}, \text{root}\}$ and the sampling probabilities $P(+) = \frac{1}{2}$ and $P(1) = \dots = P(9) = \frac{1}{18}$, which are changed to $P(1) = \dots = P(9) = \frac{1}{9}$ once 3 plus operators are generated.

Once the initial graph is generated, we apply Peano's addition axiom until the addition is resolved. In more detail, we apply the following rules.

1. $F + 0 = F$ for any formula $F$. Accordingly, if we find two nodes $i, j$ with $\vec{x}_i = +$ and $\vec{x}_j = 0$ such that $j$ is the second child of $i$, we impose $\hat{\nu}_i = \hat{\nu}_j = -1$.

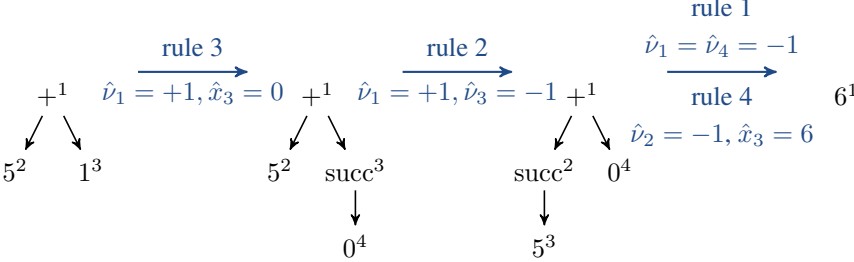

Figure 11: An example time series from the Peano dataset. The initial formula on the left is evaluated until only a single number - namely the result of the addition - is left. Exponents indicate the index of each node in the graph and blue arrow labels indicate the teaching protocol.

2. $F + \mathrm{succ}(G) = \mathrm{succ}(F) + G$ for any formulae $F$ and $G$. Accordingly, if we find two nodes $i, j$ with $\vec{x}_i = +$ and $\vec{x}_j = \mathrm{succ}$ such that $j$ is the second child of $i$, we impose $\hat{\nu}_i = +1$, $\hat{y}_i = \mathrm{succ}$, and $\hat{\nu}_j = -1$. Again, notice that we use tree edits (Zhang & Shasha, 1989) instead of graph edits in this case, such that an insertion at node $i$ automatically establishes an edge between $i$ and the newly inserted node as well as between the newly inserted node and the first child of $i$, while cutting the direct connection between $i$ and its first child, *i.e.* the new child is inserted between $i$ and its current first child.

3. $F + n = F + \mathrm{succ}(n - 1)$ for any number $n \in \{1, \ldots, 9\}$. Accordingly, if we find two nodes $i, j$ with $\vec{x}_i = +$ and $\vec{x}_j \in \{1, \ldots, 9\}$, such that $j$ is the second child of $i$, we impose $\hat{\nu}_i = +1$, $\hat{y}_i = \mathrm{succ}$, and $\hat{y}_j = n - 1$ in one-hot coding.

4. $\mathrm{succ}(n) = n + 1$ for any number $n \in \{0, \ldots, 9\}$. If $n = 9$, we define $n + 1 = 0$ to remain in the alphabet. Accordingly, if we find two nodes $i, j$ with $\vec{x}_i = \mathrm{succ}$, $\vec{x}_j \in \{0, \ldots, 9\}$, and $(i, j) \in E$, we impose $\nu_i = -1$ and $\hat{y}_j = n + 1$.

The next step in the dynamic is the graph that results from applying all edits imposed by the teaching protocol. The dynamic process ends once no rules apply anymore. An example time series is shown in Figure 11.

Note that, for the Peano dataset, the order of children is important. Accordingly, we assume that there is an auxiliary attribute that codes whether the node is a second child (where the attribute has value 1) or not (value 0). Further, let $\phi(x)$ denote the one-hot coding of symbol $x \in \Sigma$ and let $\phi_{II}$ denote the vector that is zero except for a one at the auxiliary attribute. Then, we use the following setting of $f_{\mathrm{aggr}}^1$, $f_{\mathrm{merge}}^1$, $f_{\mathrm{aggr}}^2$, and $f_{\mathrm{merge}}^2$ from Equation 1.

$$f_{\mathrm{aggr}}^1(\mathcal{N}^+, \mathcal{N}^-) = 0, \quad \text{and} \qquad f_{\mathrm{merge}}^1(\vec{x}, 0) = \mathrm{ReLU}\begin{pmatrix} \phi(+)^T \cdot \vec{x} \\ \phi(\mathrm{succ})^T \cdot \vec{x} + \phi_{II}^T \cdot \vec{x} - 1 \end{pmatrix}$$

$$f_{\mathrm{aggr}}^2(\mathcal{N}^+, \mathcal{N}^-) = \begin{pmatrix} \sum_{j \in \mathcal{N}^+} \phi^1(j) \\ \sum_{j \in \mathcal{N}^-} \phi^1(j) \end{pmatrix} \quad \text{and} \quad f_{\mathrm{merge}}^2\left(\vec{x}, \begin{pmatrix} \vec{y} \\ \vec{z} \end{pmatrix}\right) = \mathrm{ReLU}\begin{pmatrix} x_1 + y_2 - 1 \\ x_2 + z_1 - 1 \end{pmatrix}$$

Again, we distinguish between the neighborhood $\mathcal{N}^+$ of children and $\mathcal{N}^-$ of parents. The resulting node representation in layer 2 is as follows. $\phi_1^2(i) = 1$ if $\vec{x}_i = +$ and $i$ has a second child $j$ with $\vec{x}_j = \mathrm{succ}$, otherwise $\phi_1^2(i) = 0$. Further, $\phi_2^2(i) = 1$ if $\vec{x}_i = \mathrm{succ}$ and $i$ has a parent $j$ with $\vec{x}_j = +$. Accordingly, we can implement rule 2 by setting $\nu_i = \phi_1^2(i) - \phi_2^2(i)$ and $\vec{y}_i = \phi(\mathrm{succ})$ if $\phi_1^2(i) = 1$.

Table 3: The average precision and recall values ($\pm$ std.) across five repeats fro all edit types on the max degree dataset.

| | node insertion | | node deletion | | edge insertion | | edge deletion | |
| --- | --- | --- | --- | --- | --- | --- | --- | --- |
| model | recall | precision | recall | precision | recall | precision | recall | precision |
| VGAE | $1.00 \pm 0.0$ | $1.00 \pm 0.0$ | $0.35 \pm 0.2$ | $0.51 \pm 0.2$ | $1.00 \pm 0.0$ | $0.83 \pm 0.1$ | $1.00 \pm 0.0$ | $0.60 \pm 0.3$ |
| XE-GEN | $1.00 \pm 0.0$ | $1.00 \pm 0.0$ | $0.49 \pm 0.3$ | $1.00 \pm 0.0$ | $1.00 \pm 0.0$ | $1.00 \pm 0.0$ | $1.00 \pm 0.0$ | $1.00 \pm 0.0$ |
| GEN | $1.00 \pm 0.0$ | $1.00 \pm 0.0$ | $0.99 \pm 0.0$ | $1.00 \pm 0.0$ | $1.00 \pm 0.0$ | $1.00 \pm 0.0$ | $1.00 \pm 0.0$ | $1.00 \pm 0.0$ |

# E   MAX DEGREE EXPERIMENT

In addition to the graph dynamical systems presented in the main part of the paper, we also investigated whether our network is able to identify the node with maximum degree in a graph and delete it. In particular, we generated random graphs from the Barabási–Albert model with a random number of nodes sampled uniformly from $\{4, 5, 6, 7, 8\}$ and 3 new edges per iteration. We then added a special node to the graph that is connected to all other nodes to facilitate communication across the graph. In this setup, identifying the node with maximum degree should be possible in three graph neural network layers: The first layer computes the degree for each node, the second layer communicates the degree information to the special node, and the third layer compares the degree of every node to the information stored at the special node, thus identifying the node with maximum degree. Accordingly, we used graph neural networks with 3 layers and 64 neurons in each layer. All other training hyperparameters were chosen as in the other experiments.

The results are shown in Table 3. We observe that VGAE does not manage to identify the node with maximum degree, whereas the GEN with hinge loss is. Interestingly, the crossentropy loss GEN is not able to identify the node either (as is visible in the $0.49$ value for node deletion recall). This may be because the crossentropy loss converges to zero slower and thus nodes that are correctly classified can still influence the gradient, whereas the hinge loss sets the gradient for nodes that are a margin of safety away from the decision boundary to zero (refer to Section B).

# F   ADDITIONAL EXPERIMENTS ON SYNTAX TREES

We additionally evaluated graph edit networks on six datasets of Python syntax trees, where students iteratively solve a Python programming task, in particular implementing the mathematical function $f(x) = x^4 - x^2 + \frac{x}{4}$ in Python (fun), plotting the function (plt), implementing its gradient (grad), implementing gradient descent on it (desc), and finding optimal starting values (fin), as well as writing a sorting program (pysort). While the latter dataset is synthetic, the former five are recordings of fifteen students. In all cases, our task is to predict the next state of the student's program, represented by its abstract syntax tree. For all datasets we used a graph edit network with four layers, 128 neurons per layer, residual connections, and $\texttt{tanh}$ nonlinearity, which we optimized using Adam with a learning rate of $10^{-3}$ and weight decay of $10^{-3}$ for 10k epochs. In each epoch we computed the loss for one randomly sampled time series. To obtain statistics we performed a 5-fold crossvalidation on all datasets.

Table 4 shows the average root mean square tree edit distance between the predicted tree and the actual next syntax tree on all datasets, for graph edit networks (GENs, last row), as well as two baselines, namely the constant prediction (const.), *i.e.* predicting no change, and the Gaussian process scheme (GP) of Paaßen et al. (2018). We observe that both the synthetic syntax trees as well as the actual student data are hard to predict, as neither GP nor GEN outperforms the constant baseline. Further, we observe that GEN performs slightly better than GP on plt, grad, and desc, clearly better on pysort (by more than four standard deviations) and slightly worse on fun and fin. Overall, we can not observe a clear advantage of GEN on these data in terms of RMSE, which is likely due to the small set of training data (only 15 different time series). Still, in terms of inference time GENs (below 100 ms on all datasets) clearly outperform the expensive kernel-to-tree translation approach of Paaßen et al. (2018) (ranging from below 100 ms to over 2 minutes per prediction on pysort).

Table 4: The mean RMSEs ($\pm$ std.) for the constant baseline, Gaussian process regression (GP) of Paaßen et al. (2018), and graph edit networks on all syntax tree datasets.

| model | fun | plt | grad | desc | fin | pysort |
|---|---|---|---|---|---|---|
| const. | $7.95 \pm 1.1$ | $3.78 \pm 1.2$ | $6.85 \pm 2.3$ | $3.97 \pm 0.6$ | $1.45 \pm 0.4$ | $13.45 \pm 1.3$ |
| GP | $9.69 \pm 1.4$ | $6.61 \pm 2.6$ | $8.70 \pm 1.7$ | $13.97 \pm 3.8$ | $1.32 \pm 0.5$ | $26.53 \pm 1.8$ |
| GEN | $11.68 \pm 1.4$ | $5.66 \pm 2.1$ | $8.23 \pm 2.1$ | $9.10 \pm 3.4$ | $1.70 \pm 1.0$ | $18.89 \pm 1.2$ |

## G  PROMOTING SHORTER EDIT PATHS IN GEN'S TRAINING

Denote with $\mathcal{C}(\mathrm{del}), \mathcal{C}(\mathrm{ins}), \mathcal{C}(\mathrm{edel})$ and $\mathcal{C}(\mathrm{eins})$ the edit costs associated with node deletion, node insertion, edge deletion and edge insertion. Define

$$
\ell_{sp}(G) = \left( \|(\mathbf{1} - \boldsymbol{\mathcal{E}}) \odot \boldsymbol{A}\|_2 + \left\| \boldsymbol{A}(\vec{1} - \vec{\nu}) \right\|_2 \right) \, \mathcal{C}(\mathrm{edel})
$$
$$
+ \left( \|\boldsymbol{\mathcal{E}} \odot (\mathbf{1} - \boldsymbol{A})\|_2 + \|\vec{\nu}\|_2 \right) \, \mathcal{C}(\mathrm{eins}) + \left\| \vec{1} - \vec{\nu} \right\|_2 \mathcal{C}(\mathrm{del}) + \|\vec{\nu}\|_2 \mathcal{C}(\mathrm{ins})
$$

where $\vec{1}, \mathbf{1}$ are a $n$-dimensional vector and $n \times n$ matrix of all ones.

Assuming the edge and node scores have been passed through a squashing function in $(0, 1)$, then adding term $\ell_{sp}$ to the GEN's training loss will penalize costly edit paths, hence promoting shorter ones.

The total cost derived by constructing edits in $\boldsymbol{\mathcal{E}}$ is, in fact, given by

$$
|\{\mathrm{edel}_{ij} \in \boldsymbol{\mathcal{E}}\}| \, \mathcal{C}(\mathrm{edel}) + |\{\mathrm{eins}_{ij} \in \boldsymbol{\mathcal{E}}\}| \, \mathcal{C}(\mathrm{eins})
$$
$$
= |\{(i,j) \in [n]^2 : \epsilon_{ij} < -\tfrac{1}{2}, \nu_i, \nu_j > -\tfrac{1}{2}, a(G)_{ij} = 1\}| \, \mathcal{C}(\mathrm{edel})
$$
$$
+ |\{(i,j) \in [n]^2 : \epsilon_{ij} > \tfrac{1}{2}, \nu_i, \nu_j > -\tfrac{1}{2}, a(G)_{ij} = 0\}| \, \mathcal{C}(\mathrm{eins}).
$$

We can rewrite it in matrix form. Define for convenience

$$
\boldsymbol{\mathcal{E}}_- = \boldsymbol{\mathcal{E}} < -\frac{1}{2}, \quad \boldsymbol{\mathcal{E}}_+ = \boldsymbol{\mathcal{E}} > \frac{1}{2}, \quad \vec{\nu}_- = \vec{\nu} < -\frac{1}{2}, \quad \vec{\nu}_+ = \vec{\nu} > \frac{1}{2}.
$$

where the operators $>$ and $<$ are applied component wise and return 1 if "true" and 0 if "false". The matrix form becomes

$$
\left\| \boldsymbol{\mathcal{E}}_- \odot (\vec{1} - \vec{\nu}_-)(\vec{1} - \vec{\nu}_-)^\top \odot \boldsymbol{A} \right\|_0 \mathcal{C}(\mathrm{edel}) + \left\| \boldsymbol{\mathcal{E}}_+ \odot (\vec{1} - \vec{\nu}_-)(\vec{1} - \vec{\nu}_-)^\top \odot (\mathbf{1} - \boldsymbol{A}) \right\|_0 \mathcal{C}(\mathrm{eins})
$$

where $\odot$ is the element-wise product.

Conversely, the cost associated with vector $\vec{\nu}$ is

$$
\sum_{v:\nu_v < -\frac{1}{2}}^{n} \left( |\{u \in [n] : a(G)_{uv} = 1\}| \, \mathcal{C}(\mathrm{edel}) + \mathcal{C}(\mathrm{del}) \right) + \sum_{v:\nu_v > \frac{1}{2}}^{n} \left( \mathcal{C}(\mathrm{eins}) + \mathcal{C}(\mathrm{ins}) \right)
$$
$$
= \vec{1}^\top \boldsymbol{A}(G) \vec{\nu}_- \, \mathcal{C}(\mathrm{edel}) + \vec{1}^\top \vec{\nu}_- \, \mathcal{C}(\mathrm{del}) + \vec{1}^\top \vec{\nu}_+ (\mathcal{C}(\mathrm{eins}) + \mathcal{C}(\mathrm{ins})).
$$

Assuming that both edge and node scores are bounded by 1, adding the following term to the loss will promote shorter edit path

$$
\ell_{sp}(G) = \|(\mathbf{1} - \boldsymbol{\mathcal{E}}) \odot \boldsymbol{A}\|_2 \mathcal{C}(\mathrm{edel}) + \|\boldsymbol{\mathcal{E}} \odot (\mathbf{1} - \boldsymbol{A})\|_2 \mathcal{C}(\mathrm{eins})
$$
$$
+ \left\| \boldsymbol{A}(G)(\vec{1} - \vec{\nu}) \right\|_2 \mathcal{C}(\mathrm{edel}) + \left\| (\vec{1} - \vec{\nu}) \right\|_2 \mathcal{C}(\mathrm{del}) + \|\vec{\nu}\|_2 (\mathcal{C}(\mathrm{eins}) + \mathcal{C}(\mathrm{ins})).
$$

