# OpenReview forum: "Graph Edit Networks"
_ICLR.cc/2021/Conference — ICLR 2021 Poster_

### Official Review · AnonReviewer3 · 2020-10-24
**A new tool for graph editing**

**Rating:** 7
**Confidence:** 1

**Review:**

Graph Editing is a prominent research area which overlaps a variety of fields in computer science. As a typical example, a series of graphs - each obtained from its predecessor - can naturally represent the evolution of a system over time. From this viewpoint, it would be natural to obtain a means of predicting how such a series of graphs will evolve over time. The authors introduce a simple output layer (called "GEN") that can be used in graph neural networks to do precisely that.

On the theoretical side, the authors show that GENs have several useful properties, such as the ability to provide a constant-factor approximation of the Graph Edit Distance problem (assuming that a suitable teaching signal is provided). On the empirical side, the authors demonstrate the capabilities of GENs via several experimental evaluations, comparing the results to those of existing time series prediction approaches such as variational graph autoencoders. The experimental results are mostly encouraging: unlike existing systems, GENs can predict node insertions, and GENs also achieved higher accuracy for predicting edge operations.

The paper is well written. While the introduction and definitions are aimed mostly at researchers who have some knowledge on the topic (and hence I could not verify the details of the results), I believe that the paper would be of interest to the ICLR community. From what I have seen and understood, I think the contribution is also sufficient to warrant a presentation at the conference.

Questions:
-In Theorem 1, isn't it important to clarify (in the statement) that the translation can be carried out in polynomial time?
-Is there a specific reason for using the "Edit Cycles" and "Degree Rules" systems in the experiments? Both seem like rather arbitrary choices. Would it be possible to use GENs for systems where, e.g., we simply delete an edge incident to a highest-degree vertex in the graph? Or the highest-degree vertex?
-I'm curious whether GENs could be used to develop a neural network capable of learning to solve classical problems such as computing Treewidth... that problem can also be formulated as a graph editing task (via the elimination ordering characterization).

Minor remark:
Page 6: "GENs capability..." should maybe be "the capability of GENs..."; also, "Experiments are reported in three groups cover" should be "Experiments are reported in three groups and cover"

Post-Rebuttal Update:
I acknowledge having read the author's response and I also glanced over the new experiments.

---

### Official Review · AnonReviewer1 · 2020-10-29
**Nice idea, useful approach, Low technical novelty**

**Rating:** 7
**Confidence:** 2

**Review:**

The authors consider Graphical Neural Networks in the time series model: a sequence of "edits" (in its general form) describe the difference between two graphs. This approach generalizes over previous works where changes on node attributes or addition and deletion of lone edges were considered.
The advantages of this work are more pronounced in applications where the output of the neural network can be encoded as a number of graph edits (this number being much smaller than the size of the whole graph). This is well motivated.
The only weakness of this work is its technical novelty. I think all the ingredients of the work were already available. Though I like the combination as well as their theoretical analysis.
I also think the paper is relevant enough to be cited:
@inproceedings{li2019graph,
  title={Graph Matching Networks for Learning the Similarity of Graph Structured Objects},
  author={Li, Yujia and Gu, Chenjie and Dullien, Thomas and Vinyals, Oriol and Kohli, Pushmeet},
  booktitle={International Conference on Machine Learning},
  pages={3835--3845},
  year={2019}
}

---

### Official Review · AnonReviewer4 · 2020-10-29
**Official Blind Review**

**Rating:** 6
**Confidence:** 5

**Review:**

Summary
-------
This paper proposes a model that, given one graph, predicts a sequence of edits that transforms this graph to the next one in a sequence of evolving graphs. To this end, it proposes the graph edit network (GEN), which is a linear output layer that transforms node embeddings to a set of scores that are then used to deterministically select graph edits. The space of edits predictable by this model covers the full space of graph changes. The paper moreover shows that, given a graph matching and a pair of graphs, we can algorithmically find a near-optimal graph edit sequence for generating training data. Finally, the paper demonstrates the model's capabilities on a set of synthetic benchmarks.

Strong points
------------------

+ The paper is well written and properly embedded in the literature.
+ The approach of solving graph transformation by predicting a complete sequence of graph edits (i.e. including deletion) is novel and certainly very interesting.

Weak points
------------------

- I am not really convinced by the approach. There is no way for the proposed model to know if it is predicting step K+1 of the ongoing edit sequence or step 1 of a new sequence (put differently: the model has no memory/is stateless; unlike to e.g. existing recurrent graph generative models). There is also no signal telling the user that the edit sequence is finished. So it seems to me that neither the model nor the user can know which edit steps are actually predicted graphs and which are just intermediate states.
- The experiments only use synthetic toy datasets and a single baseline each. There are no comparisons with recently published models using the common, but supposedly inferior method of generating the next graph from scratch.
- Theorem 1 does not "prove that graph edit networks can approximate the NP-hard graph edit distance" or show that "edits predicted by the GEN can be sharply bounded to be close to optimal", as (incorrectly) claimed in the introduction/abstract. The theorem merely shows that the space of possible edits is complete and that the edit sequence length of your training data can be bounded if you have an (NP-hard) optimal graph matching. There is no guarantee whatsoever on what the GEN might predict. Also, GEN doesn't use any mapping, so the statement "provided that an optimal mapping is used" is rather confusing.
- Theorem 1 merely transforms the graph edit problem into a graph matching problem. This is stated on page 5, but considering that graph matching is still NP-hard, the importance of Theorem 1 seems to be overstated throughout the rest of the paper. Especially since it is largely just a variant of Proposition 1 by Bougleux et al. 2017 (as cited).
- The hinge loss with margin is rather common in deep learning, see e.g. [1]. The novelty of the proposed loss function is thus significantly overstated. [Side note: Its description on page 6 also seems overly complicated. I would recommend that you simplify this.]
- The paper states that the model can achieve linear instead of quadratic runtime if the number of edge edits are limited. While this might be true, there is no measure on how such a constraint would affect model performance. The experiment described in the last paragraph of the experimental section only provides some data on scaling behavior, not on accuracy. Especially considering that this restriction limits the number of edits by the model and thus breaks its universality it can severely affect the outcome. [Side note: The number of non-negative edge filter scores actually does not need to be constant, it merely needs to scale with sqrt(N) instead of N. So you actually have more wiggle room than you allow yourself.]

Recommendation
--------------
I recommend rejection, mainly because of significantly lacking experiments and overstated theoretical results. The presented theoretical results are interesting but do not really give deep insights about proposed approach (specifically Theorem 1). The overall theoretical contribution is significantly smaller than the authors make them out to be.

Feedback for the authors
------------------------
I really appreciate the novel approach of predicting a complete instead of a limited edit sequence. I would recommend acceptance if you a) significantly extend the experiments to include real (ideally practically relevant) graph sequences and directly compare to multiple relevant methods, which you already mention in the related work, b) solve the stopping problem mentioned above (i.e. distinguishing between another and the first edit step), and c) make your statements more accurate and avoid overstating your results. Since especially points a) and b) are unlikely possible to be solved within 2 weeks I strongly encourage you to build on this work and submit a better version at the next conference.

[1] Junbo Zhao, Michael Mathieu, and Yann LeCun. Energy-based generative adversarial network. ICLR 2017


[Update after author responses]: The authors have addressed most of my concerns and I've updated the score accordingly.

---

### Official Review · AnonReviewer2 · 2020-11-02
**Graph edit networks deal with dynamic graphs but there is probably a problem with the description of the algorithm**

**Rating:** 3
**Confidence:** 4

**Review:**

This paper proposes an algorithm to learn dynamics on graphs: node scores are computed by a GNN and then used to predict the modification of the graph (node/edge insertion/deletion).

The training of the GNN is unclear to me. In particular, the authors do not describe how they obtain the 'teaching signal'. Theorem 1 is of little help as 'graph mapping' are not defined.

I have a fundamental problem with this paper: it looks like the GNN is trained to mimic the teaching signal but as claimed by the authors on page 5 '...the key theoretical result of our paper yields a simple way to construct teaching signals...'. Hence what is the point of using a GNN if a simple algorithm gives a satisfactory answer?

I do not agree with the authors about the expressive power of GEN: for simplicity, consider a cycle graph, then for a GNN as the one described in (2), the node representation will all be the same. Hence the node scores will be the same for all nodes and the algorithm 1 will have the same output for all nodes and edges. What am I missing?

[After rebuttal]  Thanks for the clarification, the aim of the authors becomes clearer. However I still think the paper requires more work before publication.
Your definition of graph mapping is still unclear. From your definition, it looks like the only dependence of $\Psi$ with respect to the graphs $G_t$ and $G_{t+1}$ are through their number of vertices N and M: " bijective mappingψ:{1,...,M+N} → {1,...,M+N}with  the  additional  restriction  that  for  the  setInsψ:={j≤N|ψ−1(j)> M}we  obtainψ−1(Insψ) ={M+ 1,...,M+|Insψ|}"
How can this mapping be related to the edit distance of $G_t$ and $G_{t+1}$.
After reading the other reviews, I think the authors should clarify if their gaph mapping is related to the standard graph matching pb see https://en.wikipedia.org/wiki/Graph_matching.

---

### Decision · Program_Chairs · 2021-01-07
**Final Decision**

**Decision:**

Accept (Poster)

**Comment:**

The reviewers generally liked the paper but had several concerns. The rebuttal and revision of the paper could mitigate most concerns and the reviewers are now mostly positive towards the paper. Remaining concerns are mostly about the presentation of the paper which indeed has room for improvements but overall is good enough to accept the paper.